# Unveiling AI's Blind Spots: An Oracle for In-Domain, Out-of-Domain, and Adversarial Errors

## Abstract

AI models make mistakes when recognizing images—whether in-domain, out-of-domain, or adversarial. Predicting these errors is critical for improving system reliability, reducing costly mistakes, and enabling proactive corrections in real-world applications such as healthcare, finance, and autonomous systems. However, understanding what mistakes AI models make, why they occur, and how to predict them remains an open challenge. Here, we conduct comprehensive empirical evaluations using a "mentor" model —a deep neural network designed to predict another model's errors. Our findings show that the mentor model excels at learning from a mentee's mistakes on adversarial images with small perturbations and generalizes effectively to predict in-domain and out-of-domain errors of the mentee. Additionally, transformer-based mentor models excel at predicting errors across various mentee architectures. Subsequently, we draw insights from these observations and develop an "oracle" mentor model, dubbed SuperMentor, that achieves 78% accuracy in predicting errors across different error types. Our error prediction framework paves the way for future research on anticipating and correcting AI model behaviours, ultimately increasing trust in AI systems. All code, models, and data will be made publicly available.

## 1 Introduction

AI models are prone to making errors in image recognition tasks, whether dealing with in-domain, out-of-domain (OOD), or adversarial examples. In-domain errors occur when models misclassify familiar data within the training domain, while OOD errors arise when faced with unseen or out-of-domain data. Adversarial errors are particularly concerning, as they result from carefully crafted perturbations designed to mislead the model.

Accurately predicting these errors is critical to enhancing the overall robustness and reliability of AI systems, especially in high-stakes real-world applications such as healthcare (Habehh & Gohel, 2021), finance (Mashrur et al., 2020), and autonomous driving (Huang et al., 2022). Proactively identifying potential errors enables more efficient corrections, reducing costly mistakes and safeguarding against catastrophic failures. By predicting when models are likely to err, we can implement strategies that either mitigate or entirely avoid the risks associated with those errors, ultimately leading to more trustworthy AI deployments.

Understanding the specific types of errors AI systems make, the reasons why they make these errors, and most importantly, how to predict these errors remains an unresolved challenge. Existing literature on error monitoring systems for AI models encompasses various approaches, including uncertainty estimation (Nado et al., 2021; Lakshminarayanan et al., 2017), anomaly detection (Bogdoll et al., 2022), outlier detection (Boukerche et al., 2020), and out-of-domain detection (Yang et al., 2024). While these methods are crucial for assessing model reliability, they mainly focus on determining whether a given data point falls outside the scope of the model's training. Thus, these approaches misalign with our primary objective of predicting whether AI models will make mistakes, as models can err on familiar data while behaving correctly on out-of-scope samples.

Subsequent research in out-of-domain detection has demonstrated that a model's accuracy is often correlated with how far the data deviates from in-domain samples (Hendrycks & Dietterich, 2019;

Figure 1: **AI models make mistakes and an "oracle" mentor model predicts when they will happen.** A "mentee" neural network (black) was trained for multi-class image recognition, but it can still misclassify in-domain, out-of-domain, and adversarial images. For instance, it might mislabel an in-domain dog image as a cat. The mentor model (blue), inputting the same images as the mentee, predicts whether the mentee will make a mistake. For example, if the mentee incorrectly labels an adversarial dog image, the mentor's ground truth label is "wrong"; conversely, if the mentee correctly labels an out-of-domain dog image, the mentor's label is "correct". The mentee's parameters are frozen (snowflake), while the mentor's are trainable (fire). During inference (orange), the mentor predicts whether the mentee will make an error on test images that have never been seen by both the mentee and the mentor.

Shankar et al., 2021; Li et al., 2017). These methods typically rely on predefined metrics, such as model parameter distances (Yu et al., 2022), model disagreements (Jiang et al., 2021; Madani et al., 2004) and confidence scores (Guillory et al., 2021), which limits their ability to generalize predictions across various data types, including errors arising from in-domain data or adversarial attacks (Szegedy, 2013). Another line of research improves the robustness of the AI models with adversarial training approaches(Ilyas et al., 2019; Gowal et al., 2020; Balunović & Vechev, 2020); however, these approaches primarily focus on improving the model's overall performance rather than predicting when errors may occur in the models.

Different from all these previous works, we delve into the underlying principles of errors generated by AI models in the task of image classification with another AI model. Specifically, we designate the AI model that predicts errors as the **mentor** and the AI model being evaluated for performance as the **mentee**. The mentor strives to predict whether the mentee makes a mistake for any given test data. See **Fig. 1** for the detailed illustration of the problem setup. Training the mentor on the error patterns made by the mentee can potentially reveal the strengths and weaknesses of the mentee's learned representations across various visual contexts.

Specifically, we examine the effects of three distinct error types AI models often make: In-Domain (ID) Errors, Out-of-Domain (OOD) Errors, and Adversarial Attack (AA) Errors on three increasingly complex image datasets CIFAR-10 (Krizhevsky et al., 2009), CIFAR-100 (Krizhevsky et al., 2009) and ImageNet-1K (Deng et al., 2009). We identify which of these error types has the most significant impact on the mentor's error prediction performances, and explore the reasons behind its prominence. Additionally, we assess how different mentor architectures influence error prediction accuracy and evaluate the mentor's generalization performance across various mentee architectures. Finally, we develop a SuperMentor model that successfully predicts errors of the mentee with 78% accuracy across diverse error types. Our main contributions are highlighted below:

**1.** We conduct an in-depth analysis of how training mentors on each of three distinct error types specified by the mentees—In-Domain (ID) Errors, Out-of-Domain (OOD) Errors, and Adversarial Attack (AA) Errors—affect the performance of error predictions over three increasingly complex image datasets. Our results reveal that training mentors with adversarial attack errors from the mentee has the most significant impact on improving the mentor's error prediction accuracy.

**2.** We explore how various mentor model architectures affect error prediction performance. Our experiments demonstrate that transformer-based mentor models outperform other architectures in accurately predicting errors.

**3.** We investigate how varying levels of distortion in OOD and adversarial images affect the accuracy of error predictions. The findings indicate that training mentors with images with small perturbations

can improve error prediction accuracy. In addition, we show that a mentor trained to learn error patterns from one mentee can successfully generalize its error predictions to another mentee.

**4.** Based on our findings from points 1 to 3, we present the SuperMentor model, which predicts errors across diverse mentee architectures and error types. Experimental results show that SuperMentor outperforms baseline mentors, demonstrating its superior error-predictive capabilities.

## 2 RELATED WORK

**Error monitoring systems for AI models.** With the growing deployment of AI models across diverse fields, ensuring their reliability and understanding their limitations has become increasingly crucial. This has led to numerous research in safe AI such as uncertainty estimation (Nado et al., 2021; Lakshminarayanan et al., 2017), anomaly detection (Bogdoll et al., 2022), outlier detection (Boukerche et al., 2020), and out-of-domain detection (Yang et al., 2024). Unlike these areas, which mainly aim to predict whether the input data falls outside the training domain, our focus is on monitoring and predicting errors in AI models by determining whether the model's output is correct, irrespective of whether the data comes from the training domain.

Moreover, to detect whether the input data is out of scope, the prior approaches mainly rely on softmax outputs (Granese et al., 2021; Hendrycks & Gimpel, 2016; Dang et al., 2024) or activations from network layers (Wang et al., 2020; Cheng et al., 2019; Ferreira et al., 2023), in applications such as object detection (Kang et al., 2018) and trajectory prediction (Shao et al., 2023; 2024). However, these methods often depend on manually defined metrics to estimate the probability of a mentee making a mistake. In contrast, our approach leverages another AI model to automatically learn and approximate the mentee's decision boundaries, predicting its errors in an end-to-end trainable manner.

**Out-of-domain detection.** Our research on predicting mentee errors is closely related to out-of-domain detection in error monitoring systems, though it differs in several key aspects. As highlighted by (Guérin et al., 2023), error prediction is distinct from OOD detection (Liu et al., 2020a; Sun et al., 2021; Lee et al., 2018; Sun et al., 2022) in their objectives. While OOD detection aims to detect whether the given data comes from the same domain as the training set, the aim of error prediction is to learn whether the mentee will make a mistake on the given data. In other words, out-of-domain data may not necessarily cause the model to err, and model errors can also occur on in-domain data.

Recent studies (Hendrycks & Dietterich, 2019; Shankar et al., 2021; Li et al., 2017) have shown that a model's accuracy on a given dataset is often correlated with how far the data deviates from in-domain samples. However, these studies typically rely on pre-defined metrics, such as model parameter distances (Yu et al., 2022), model disagreements (Jiang et al., 2021; Madani et al., 2004), confidence scores (Guillory et al., 2021), domain-invariant representations (Chuang et al., 2020), and domain augmentation (Deng et al., 2021a), limiting their ability to generalize error prediction for in-domain data. In contrast, our mentor model is capable of predicting both OOD and in-domain errors for a mentee. Additionally, our mentor is an AI model trained end-to-end, eliminating the need for manually defined criteria.

**Adversarial attack and defense.** In addition to OOD error, (Szegedy, 2013) discovered that deep neural networks can be fooled using input perturbations of extremely low magnitude. Building upon this finding, a substantial number of adversarial attacks have been proposed, including white-box attacks (Goodfellow et al., 2014; Mądry et al., 2017; Carlini & Wagner, 2017; Schwinn et al., 2023; Gao et al., 2020), black-box attacks (Uesato et al., 2018; Rahmati et al., 2020; Brendel et al., 2017; Chen et al., 2020), and backdoor attacks (Liu et al., 2020b; Xie et al., 2019; Kolouri et al., 2020). To defend against these adversarial attacks, various defence mechanisms (Qin et al., 2019; Deng et al., 2021b; Liu et al., 2019) have been developed to withstand or detect adversarial inputs. Furthermore, although the primary objective of adversarial attacks is to deceive AI models, there are instances where adversarial perturbations are exploited to enhance the model performance — a technique known as adversarial training (Ilyas et al., 2019; Gowal et al., 2020; Balunović & Vechev, 2020). Unlike adversarial training, which involves using adversarial samples to train the mentee, our approach focuses on teaching mentors to learn the mentee's error patterns revealed by these adversarial attack samples.

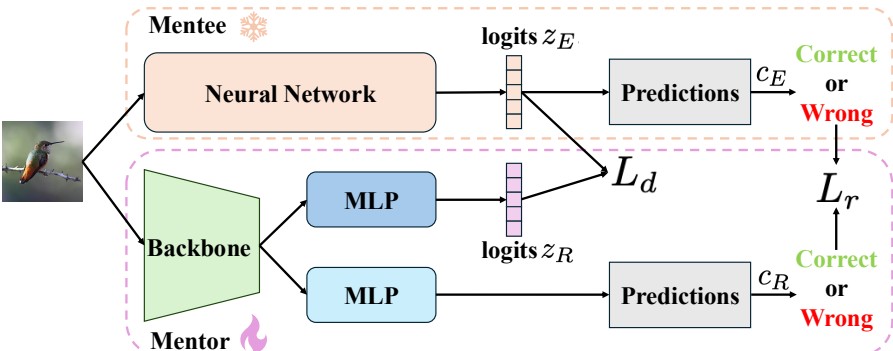

Figure 2: **Overview of a mentor model.** Given a fixed mentee model (snowflake), the mentor model takes an input image and uses a pre-trained backbone on ImageNet-1K (Deng et al., 2009) to extract features. The feature maps are then processed in two streams via multi-layer perceptrons (MLP)s. The output logits $z_R$ from one stream are compared with the mentee's output logits $z_E$ using a distillation loss $L_d$. The other stream performs a binary prediction of whether the mentee makes a mistake or not. The prediction is supervised by a logistic regression loss $L_r$. The parameters of MLPs in the two streams are not shared.

## 3 EXPERIMENTAL SETUPS

We denote the mentor and mentee networks as $f_R(\cdot)$ and $f_E(\cdot)$ respectively. We also define $\mathcal{X}$ as the domain-specific set containing all the test images for a mentee, and $\mathcal{Y}$ as their ground-truth object class labels. Therefore, a mentor is expected to make perfect predictions about the correctness of the mentee's responses (1 for "correct" and 0 for "wrong") given any image $x$ from $\mathcal{X}$:

$$\forall x \in \mathcal{X}, f_R(x) = \begin{cases} 1, & \text{if } f_E(x) = y, \\ 0, & \text{otherwise} \end{cases} \tag{1}$$

where $y \in \mathcal{Y}$ is the ground-truth object class label of the corresponding image $x$.

### 3.1 MENTORS

**Model Architecture:** We propose mentor models, as illustrated in **Fig. 2**. Given an input image, the backbone of a mentor model extracts features from the input image. We adopt either of the two backbones for the feature extractors of mentors: a 2D Convolutional Neural Network (2D-CNN) ResNet50 (He et al., 2016) and a transformer-based ViT (Dosovitskiy, 2020). The extracted feature maps are further processed in two streams implemented as multi-layer perceptrons (MLP)s. The parameters of the MLPs in the two streams are not shared.

The first stream generates logits $z_R$ by predicting the probability distribution of a mentee over all the object classes when the mentee classifies the given image. The mentee network is kept fixed while training the mentor. Let us define the mentee's output logit as $z_E$. We introduce the distillation loss proposed by (Hinton, 2015): $L_d = Distill(z_R, z_E)$ to align $z_R$ with $z_E$. We set the temperature hyper-parameter in $L_d$ as 1.0, which controls the smoothness of the soft probability distribution. Higher temperatures make the distribution softer and more uniform across classes.

In the second stream, the mentor is prompted to predict whether the mentee will make a mistake on the given image or not. We denote the predicted binary label as $c_R$, where 1 indicates that the mentee does not make a mistake and vice versa for 0. This prediction is supervised by $L_r = LR(c_R, c_E)$ where $LR(\cdot, \cdot)$ is the logistic regression loss and $c_E$ is the ground truth correctness label of a mentee. The overall loss is $L = L_d + L_r$.

**Training and Implementation Details:** All mentors are trained on Nvidia RTX A5000 and A6000 GPUs, utilizing the AdamW optimizer (Loshchilov & Hutter, 2017) with a cosine annealing scheduler (Loshchilov & Hutter, 2016), and an initial learning rate of $2 \times 10^{-4}$. All mentors load the weights of the feature extractor pre-trained on the ImageNet-1K dataset for 1000-way image classification tasks (Deng et al., 2009) and further fine-tune on the error prediction task. During training, images

are resized and center-cropped to $224 \times 224$ pixels. All the mentor models are trained for 40 epochs with a batch size of 512.

## 3.2 MENTEES AND THEIR DATASETS

We employ two architectures as the mentees' backbones: ResNet50 (He et al., 2016), which is a 2D-Convolutional neural network (2D-CNN), and ViT (Dosovitskiy, 2020), which is a transformer architecture based on self-attention mechanisms.

To train and test our mentees, we include three prominent image datasets of varying sizes and follow their standard data splits: CIFAR-10 (C10, (Krizhevsky et al., 2009)) with 10 object classes, CIFAR-100 with 100 object classes (C100, (Krizhevsky et al., 2009)) and ImageNet-1K with 1000 object classes (IN, (Deng et al., 2009)). Their multi-class recognition accuracy on the standard test sets of C10, C100 and IN datasets are 96.98%, 84.54%, 76.13% for the ResNet50 mentee and 97.45%, 86.51%, 81.07% for the ViT mentee respectively. The parameters of the mentees are frozen throughout all the experiments conducted on mentors.

## 3.3 DATASETS FOR TRAINING AND TESTING MENTORS

The mentor's objective is to predict whether the mentee will misclassify a given image, regardless of its source. The mentor is trained on correctly and wrongly classified images by a mentee. Next, we introduce how these images are curated and collected.

A mentee may encounter various types of errors when dealing with real-world data. To explore which error types most effectively reveal the mentee's learning patterns, we categorize errors into three types: (1) errors from in-domain test images, (2) errors from out-of-domain images, and (3) errors from adversarial images generated using adversarial attack methods. Next, we introduce these three error types in detail.

**In-Domain (ID) Errors.** occur on data that come from the same domain as the mentee's training dataset. Specifically, errors on images from the standard validation set of ImageNet-1K or the test sets of CIFAR-10 and CIFAR-100 are considered ID errors. Along with the correctly classified images from these standard test sets, we create three datasets for a mentor: **IN-ID**, **C10-ID**, and **C100-ID**, following the naming convention of [Dataset]-[Error Type].

**Out-of-domain (OOD) Errors.** refer to errors that arise when the mentee encounters data outside the training domain. To obtain OOD samples of a dataset, we adopt four types of image corruptions from (Hendrycks & Dietterich, 2019): **speckle noise (SpN)** (noise category), **Gaussian blur (GaB)** (blur category), **spatter (Spat)** (weather category), and **saturate (Sat)** (digital category). The noise levels can vary and we select level 1 for image corruptions as specified in (Hendrycks & Dietterich, 2019) by default. As noise levels increase, the distortions on OOD images become more pronounced, leading to more mistakes of a mentee.

Following the naming conventions of [Dataset]-[Error Type]-[Error Source], we collect correctly and wrongly classified OOD samples based on C10 images of a mentee and curate four datasets for a mentor: **C10-OOD-SpN**, **C10-OOD-GaB**, **C10-OOD-Spat** and **C10-OOD-Sat**. Without the loss of generality, we can also curate four datasets each for a mentor based on C100 and IN images of a mentee.

**Adversarial Attack (AA) Errors.** Errors from adversarial images are specifically generated by adversarial attack methods to mislead or confuse the mentee. Given our assumption that the mentor has full access to the student model's parameters, we focus exclusively on white-box adversarial attacks as they typically produce more subtle yet effective perturbations compared to their black-box counterparts. To generate adversarial images, we employ four untargeted adversarial attack methods: **PGD** (Mądry et al., 2017) creates adversarial examples by repeatedly taking steps along the loss gradient; **CW** (Carlini & Wagner, 2017) attempts to minimize the $L_2$ norm of the perturbation while ensuring misclassification. **Jitter** (Schwinn et al., 2023) adds Gaussian noise to the output logits to encourage a diverse set of target classes for the attack. **PIFGSM** (Gao et al., 2020) crafts patch-wise noise instead of pixel-wise noise. We set $c = 1.0$ in the CW attack, and perturbation bound $\epsilon = \frac{1}{255}$ for other attacks by default. See their papers for these hyper-parameter definitions. Intuitively, the attacks are stronger with higher hyper-parameter values; hence, the mentees make more mistakes.

| Mentee | Error Source | | CIFAR-10 | | CIFAR-100 | | ImageNet-1K | |
|--------|------|------|----------|------|-----------|------|-------------|------|
| | | | $N_{train}$ | $N_{test}$ | $N_{train}$ | $N_{test}$ | $N_{train}$ | $N_{test}$ |
| ResNet50 | | ID | 151/9547 | 151/151 | 773/7681 | 773/773 | 5967/32099 | 5967/5967 |
| | OOD | SpN | 690/7930 | 690/690 | 1889/4333 | 1889/1889 | 9984/20048 | 9984/9984 |
| | | GaB | 149/9553 | 149/149 | 760/7720 | 760/760 | 8013/25963 | 8012/8012 |
| | | Spat | 222/9336 | 221/221 | 990/7032 | 989/989 | 7042/28874 | 7042/7042 |
| | | Sat | 240/9282 | 239/239 | 1309/6072 | 1309/1309 | 8187/25439 | 8187/8187 |
| | AA | Jitter | 338/8988 | 337/337 | 1054/6840 | 1053/1053 | 7591/27227 | 7591/7591 |
| | | PGD | 447/8661 | 446/446 | 1180/6460 | 1180/1180 | 9009/22973 | 9009/9009 |
| | | CW | 487/8539 | 487/487 | 1120/6642 | 1119/1119 | 8102/25694 | 8102/8102 |
| | | PIFGSM | 1613/5161 | 1613/1613 | 2090/3732 | 2089/2089 | 11226/16322 | 11226/11226 |
| ViT | | ID | 128/9618 | 127/127 | 675/7977 | 674/674 | 4733/35801 | 4733/4733 |
| | OOD | SpN | 286/9144 | 285/285 | 1155/6535 | 1155/1155 | 6019/31945 | 6018/6018 |
| | | GaB | 130/9610 | 130/130 | 678/7966 | 678/678 | 6402/30794 | 6402/6402 |
| | | Spat | 170/9490 | 170/170 | 809/7573 | 809/809 | 5351/33947 | 5351/5351 |
| | | Sat | 227/9319 | 227/227 | 1219/6345 | 1218/1218 | 5883/32351 | 5883/5883 |
| | AA | Jitter | 552/8344 | 552/552 | 1232/6304 | 1232/1232 | 10325/19025 | 10325/10325 |
| | | PGD | 649/8053 | 649/649 | 1410/5770 | 1410/1410 | 14960/11680 | 11680/11680 |
| | | CW | 446/8664 | 445/445 | 1136/6592 | 1136/1136 | 8614/24158 | 8614/8614 |
| | | PIFGSM | 799/7605 | 798/798 | 1812/4564 | 1812/1812 | 15038/11654 | 11654/11654 |

Table 1: **Dataset split for each error source used in mentor training.** If the mentor is trained on the mentee's performance (ResNet50 or ViT) for a specific error source, the data in this error source will be split according to this table. $N_{train}$ and $N_{test}$ denote the number of training and testing samples, respectively, formatted as [number of samples misclassified by the mentee] / [number of samples correctly classified by the mentee].

Note that adversarial attacks are not always successful, and mentees can still correctly classify some adversarial images. We collect both the correctly and incorrectly classified adversarial images by a mentee based on C10 images, curating four datasets for the mentor: **C10-AA-PGD**, **C10-AA-CW**, **C10-AA-Jitter**, **C10-AA-PIFGSM**. Without the loss of generality, we can also curate four datasets each for a mentor based on C100 and IN images of a mentee.

**Training and Test Splits.** For any given dataset of a mentor, let $N_c$ and $N_w$ represent the sets of $n$ correctly and $m$ incorrectly classified images by a mentee. The sizes of $N_c$ and $N_w$ can vary significantly, depending on the mentee's classification performance. A mentee with high recognition accuracy will have more correct classifications (big $n$) and fewer incorrect ones (small $m$). To create a balanced test set for a mentor, we select equal numbers of correctly and incorrectly classified samples. The remaining samples are used for training. The details of the dataset split are shown in **Tab. 1**. To address the long-tail problem in the training set, during each training epoch for a mentor, we randomly generate a batch of samples that includes an equal number of correctly and incorrectly classified images by the mentee.

**Evaluation Metric.** To assess the performance of mentors, we report their error prediction accuracy on the test set corresponding to each specified error source. For instance, a mentor trained on the C10-ID training set is evaluated on the C10-OOD-SpN test set. The error prediction accuracy is calculated by averaging the mentor's accuracies on the samples that the mentee correctly classified and those that the mentee incorrectly classified. However, since a mentee can make mistakes across various real-world scenarios, a mentor must accurately predict errors across all error types. Therefore, we compute the average accuracy of a mentor across all test sets, including one ID error, four OOD errors, and four AA errors. For simplicity, we refer to this average accuracy across all nine error sources as **Accuracy**. A mentor randomly guessing whether a mentee's image classification is correct or incorrect for a given image would achieve an accuracy of 50%.

## 4 RESULTS

### 4.1 TRAINING ON SPECIFIC ERRORS OF MENTEES IMPACTS THE PERFORMANCE OF MENTORS

A mentee's mistakes can reveal their learning tendencies, behaviours, or traits. Here, we investigate which types of errors offer the most insight into understanding a mentee's decision boundaries during image recognition tasks. We train mentors with identical architectures on datasets containing specific error types made by the mentee across C10 (**Fig. 3(a)**), C100 (**Fig. 3(b)**), and IN (**Fig. 3(c)**). For instance, if a mentor trained on C10-OOD achieves higher accuracy in error prediction compared to

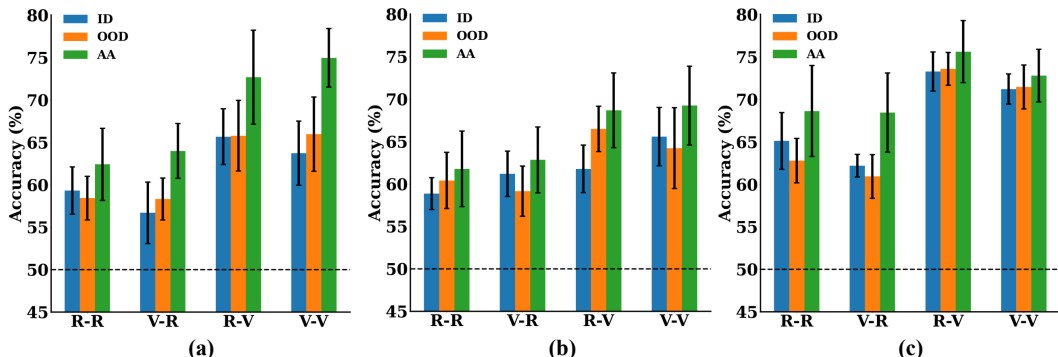

Figure 3: **Mentors trained on adversarial images of a mentee outperform mentors trained on OOD and ID images of the same mentee.** Average accuracy of a mentor trained on one type of error of a mentee for **(a)** CIFAR-10, **(b)** CIFAR-100 and **(c)** ImageNet-1K datasets is presented. Three types of errors made by a mentee are categorized based on in-domain (ID, blue), out-of-domain (OOD, orange), and images generated by adversarial attacks (AA, green). In each subplot, the labels on the x-axis are interpreted as [mentee]-[mentor], where 'V' and 'R' represent ViT and ResNet50 architectures for a mentee or a mentor respectively. Error bars indicate the standard deviation. The dotted black line indicates the chance level. See **Sec. 3.3** for error types and the evaluation metric. The four sets of bars in each subfigure correspond to the heatmaps shown in subfigures (a), (b), (c), and (d) of **Fig. S1- S3**.

one trained on C10-IN, this suggests that in-domain errors provide less diagnostic information about the mentee's decision-making process than out-of-domain errors. Both mentors and mentees may have the same or different backbones, such as ResNet50 (R) or ViT (V).

As shown in **Fig. 3**, over C10, C100, and IN images, the high accuracy for mentors trained on adversarial attack (AA) errors indicates that these errors offer deeper insights into the mentee's decision process compared to out-of-domain (OOD) and in-domain (ID) errors. In some cases, mentors trained on OOD errors slightly outperformed those trained on ID errors, though both were still inferior to those trained on AA errors.

**Loss landscape analysis.** A loss landscape of a mentee reflects how a mentee's loss function behaves across different parameter configurations. Mentors' performance offers insights into the structure of a mentee's loss landscape. Consistent with (Ilyas et al., 2019), the high accuracy of mentors trained on AA errors suggests that adversarial images lie closer to the mentee's decision boundary, enabling more accurate prediction of the mentee's mistakes and a deeper understanding of the loss landscape. Similarly, OOD data aids mentors in learning decision boundaries by shifting ID samples closer to the boundary. However, it does not explore the boundary as thoroughly as adversarial images. ID data, with fewer samples near the boundary, provides more limited exploration compared to adversarial examples.

### 4.2 MENTOR ARCHITECTURES MATTER IN ERROR PREDICTIONS

To computationally model the decision boundary of a mentee using a mentor, the mentor requires more complex architectures with a larger number of parameters than the mentee. Indeed, from **Fig. 3**, over all the datasets, we observed that utilizing ViT (V) as the mentor backbone consistently achieves higher accuracy across all error types of ViT-based and ResNet-based mentees compared to the mentor based on ResNet50 (R). One example of this performance disparity is observed in the context of the adversarial attack error type for CIFAR-10. The ViT-based mentor attains an accuracy of 74.95%, substantially higher than the accuracy of 63.99% for the ResNet-based mentor.

**Loss landscape analysis.** The performance difference between mentors' architectures is due to ViT's superior ability to identify features from error patterns. Its self-attention mechanism captures complex relationships among data samples, providing a deeper understanding of the mentee's loss landscape, particularly in modelling irregular, rugged landscapes with sharp peaks and valleys.

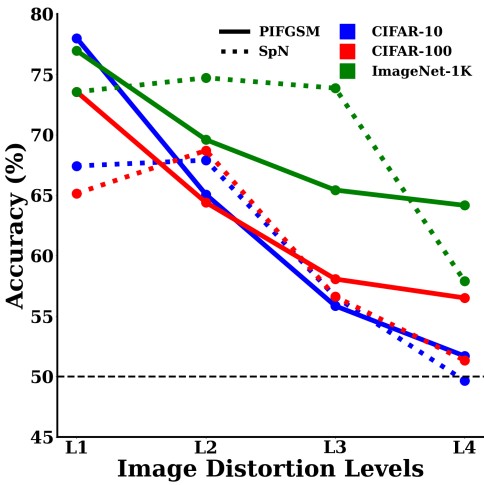

Figure 4: **A mentor's accuracy is heavily influenced by the levels of image distortions introduced by out-of-domain perturbations and adversarial attacks.** ViT mentor's accuracy is a function of varying image distortion levels from PIFGSM (Gao et al., 2020) and Speckle Noise (SpN) (Hendrycks & Dietterich, 2019) to the C10 images of a ResNet50-based mentee. The black dashed line indicates the chance level.

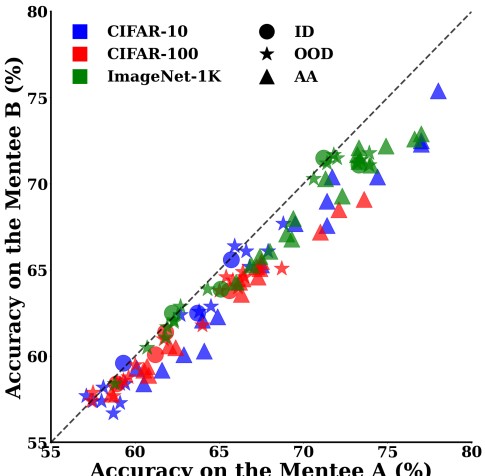

Figure 5: **Mentors can generalize their error predictions across different mentee architectures.** Mentors trained on mentee A's predictions (x-axis) are evaluated against the predictions from mentee B (y-axis). Each marker is a generalization experiment of a mentor trained on different error types (marker shapes) in different image datasets (colours) of a mentee. The black dash line indicates the diagonal.

### 4.3    TRAINING ON IMAGES WITH SMALLER PERTURBATIONS HELPS ERROR PREDICTIONS

Although adversarial images have been demonstrated to aid in error prediction (**Sec. 4.1**), it remains unclear whether adversarial images with varying degrees of image distortion exhibit the same effect. A straightforward method to regulate the level of image distortion caused by adversarial attacks is to set the perturbation bound $\epsilon$. We employ four corruption levels by setting $\epsilon = \frac{1}{255}, \frac{2}{255}, \frac{4}{255}$, and $\frac{8}{255}$. We use the adversarial attack PIFGSM as an example since the error patterns from PIFGSM are most effective for the mentor's prediction (see **Fig. S1- S3**). As shown in **Fig. 4**, the mentor's accuracy significantly decreases as the distortion level increases. In particular, for the C10-AA-PIFGSM, the accuracy at level 1 is 78.0%, which is notably higher than 51.7% at level 4. Our findings suggest that adversarial attacks employing smaller perturbations yield more benefits for mentor error prediction. This phenomenon can be attributed to the fact that adversarial images with minimal perturbations maintain closer proximity to the decision boundary of a mentee.

Building on the findings above, we investigate whether the mentor's performance is influenced by how far OOD images are from the ID data. Specifically, we aim to determine whether the degree of deviation from the training domain impacts the mentor in a similar way to our observations on adversarial images. To explore this, we analyze images corrupted with Speckle Noise (SpN) and adjust the standard deviation $\sigma$ of SpN to 0.01, 0.06, 0.15, and 0.6, representing four distinct levels of distortion. The outcomes are depicted in **Fig. 4**. We observe that the mentor's accuracy improves as the distortion introduced by SpN decreases. For example, the mentor achieves an accuracy of 67.42% on level 1 of C10-OOD-SpN, while the accuracy drops significantly to 49.66% on level 4 of C100-OOD-SpN. This suggests that OOD error types with smaller perturbations enhance the mentor's performance. However, unlike adversarial attacks, caution is necessary because the mentor's accuracy can plateau with extremely small distortion levels, as shown by the minimal difference in accuracy between levels 1 and 2 of SpN in **Fig. 4**.

### 4.4    MENTORS GENERALIZE ACROSS MENTEES

In **Sec. 4.1**, mentors have demonstrated their ability to learn the error patterns of mentees. This observation raises an important question: can the error patterns learned from one mentee (mentee A) be generalized to another mentee (mentee B) when the two mentees employ different model

Figure 6: **Our SuperMentor outperforms other mentor baselines on the CIFAR-10 dataset.** The row index follows the format [mentee]-[mentor], where 'V' and 'R' represent ViT and ResNet50 architectures for a mentee or a mentor respectively. The column index represents the error source used for training the mentor. Results in each cell denote the average error prediction accuracy over all error types with the standard deviation over 3 runs. Our SuperMentor's accuracy is highlighted in red boxes. The detailed performance of all mentors on specific error types is depicted in **Fig. S1**.

| | $L_d$ | $L_a$ | ID | OOD | | | | AA | | | | Average |
| | | | | SpN | GaB | Spat | Sat | PGD | CW | Jitter | PIFGSM | |
|---|---|---|---|---|---|---|---|---|---|---|---|---|
| | ✗ | ✗ | 57.5 | 61.0 | 56.1 | 58.6 | 54.3 | 58.6 | 59.1 | 58.5 | 59.6 | 58.2 |
| C10 | ✗ | ✓ | 80.0 | 73.7 | 79.2 | 77.9 | 74.3 | 80.5 | 76.5 | 79.7 | 71.2 | 77.0 |
| | ours | | 80.9 | 73.2 | 80.5 | 79.4 | 75.6 | 81.4 | 78.2 | 80.7 | 71.9 | **78.0** |
| | ✗ | ✗ | 56.8 | 59.5 | 56.6 | 57.8 | 53.7 | 57.7 | 57.3 | 57.3 | 57.1 | 57.1 |
| C100 | ✗ | ✓ | 75.0 | 70.9 | 74.8 | 74.1 | 68.1 | 78.1 | 75.2 | 76.2 | 66.5 | 73.2 |
| | ours | | 75.4 | 71.1 | 75.4 | 74.5 | 68.4 | 78.3 | 75.6 | 76.6 | 66.9 | **73.6** |
| | ✗ | ✗ | 73.0 | 70.1 | 69.6 | 72.8 | 68.5 | 75.8 | 72.5 | 73.6 | 70.7 | 71.9 |
| IN | ✗ | ✓ | 78.7 | 73.1 | 73.6 | 78.0 | 73.2 | 83.0 | 78.4 | 79.9 | 72.2 | 76.7 |
| | ours | | 78.9 | 73.6 | 74.6 | 78.3 | 73.6 | 83.0 | 78.4 | 79.9 | 72.3 | **77.0** |

Table 2: **Ablation study of loss components in SuperMentor.** $L_d$ denotes the distillation loss (see **Sec. 3.1**) and $L_a$ represents the alignment loss between the mentor's and mentee's predicted object class labels. SuperMentor is evaluated on the mistakes of a ResNet50-based mentee. Each result is the average of three independent runs. The performance of SuperMentor is coloured in grey. The full results with standard deviations are shown in **Tab. S4**.

architectures? To explore this, we evaluate all 324 mentors, whose performances are depicted in **Fig. S1- S3**, on the alternate mentee. Specifically, mentors trained on the errors of the ResNet50 mentee are tested on the predictions of the ViT mentee, and vice versa. The outcomes of these evaluations are illustrated in **Fig. 5**. Surprisingly, most points lie near the dashed diagonal line, implying that the mentors' performance does not significantly deteriorate when evaluated on the predictions of different mentee architectures. This finding indicates that ResNet50 and ViT mentees tend to produce similar error patterns when trained on the same dataset.

## 4.5 OUR PROPOSED SUPERMENTOR MODEL OUTPERFORMS OTHER BASELINES

By drawing insights from observations in the subsections above, we propose an "oracle" mentor model, dubbed SuperMentor. We introduce the technical novelties of our SuperMentor below. First, as demonstrated in **Sec. 4.1** and **Sec. 4.3**, mentors trained on adversarial images with small perturbations of a mentee outperform those trained on OOD and ID images; thus, our SuperMentor adopts the training data from the PIFGSM error source of mentees with $\epsilon = \frac{1}{255}$. Second, since ViT has been proven to be a more effective architecture for mentors than ResNet50 (**Sec. 4.2**), SuperMentor adopts ViT as the backbone architecture.

**Fig. 6** shows that SuperMentor outperforms other baseline mentors in the CIFAR-10 dataset. The detailed performance of the SuperMentor, along with other baseline mentors on various error sources from the CIFAR-10, CIFAR-100 and ImageNet-1K datasets, is presented in **Fig. S1- S3**.

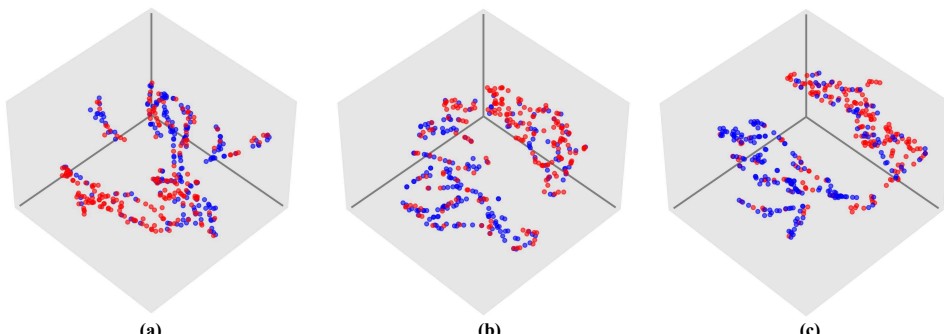

|        (a)        |        (b)        |        (c)        |

Figure 7: **3D visualization of the embeddings extracted from our SuperMentor Model for the classification of: a) C10-ID samples, b) C10-OOD-GaB samples and c) C10-AA-Jitter samples.** We use t-SNE (Van der Maaten & Hinton, 2008) to perform clusterings on the representations of our SuperMentor model for classifications of different error sources on the C-10 dataset. Red points indicate samples that the mentee fails to classify correctly, whereas blue points represent samples that the mentee successfully classifies. 200 red points and 200 blue points are randomly selected from the test sets and presented here. The visualized features are the embeddings computed based on the MLP in the second stream of the SuperMentor. Specifically, they are extracted before the final binary classification layer on whether the mentee makes a mistake.

We also present the visualization of the SuperMentor's embeddings on three types of error sources of a mentee in **Fig. 7**. It is evident that SuperMentor can effectively segregate samples correctly classified by the mentee from those that are misclassified, forming two distinct clusters.

Next, we examine the effect of the distillation loss $L_d$ (**Fig. 2**) on the SuperMentor performance. The results are presented in **Tab. 2**. It is clear that excluding $L_d$ results in a decrease in SuperMentor's accuracy across all datasets. For example, in the C10 dataset, the average accuracy of SuperMentor decreases from 78.0% to 58.2%. This suggests that $L_d$ encourages SuperMentor to learn the fine-grained decision boundaries among different object classes of a mentee.

Alternatively, instead of utilizing the mentee's logits, SuperMentor can incorporate an additional cross-entropy loss to align the mentor's predicted object class labels with those of the mentee, denoted as $L_a$. From **Tab. 2**, we observe that replacing $L_d$ with $L_a$ leads to a slight decrease in accuracy. This is due to the fact that the mentee's logits contain more information than the mentee's class labels.

## 5 CONCLUSION

In our work, we tackle the challenge of predicting errors in AI models through extensive empirical evaluations using an end-to-end trainable "mentor" model. This mentor model is designed to assess the correctness of a mentee model's predictions across three distinct error types: in-domain errors, out-of-domain errors, and adversarial attack errors. Our results show that the mentor model excels at learning from a mentee's errors on adversarial images with minimal perturbations and, surprisingly, generalizes well to both in-domain and out-of-domain predictions of the same mentee. Additionally, we highlight the effectiveness of transformer-based mentor architectures compared to 2D-CNN-based ones, demonstrating their superior generalization capabilities across mentees with diverse backbones. Lastly, we introduce the SuperMentor, which significantly outperforms all existing mentor baselines.

Our work paves the way for several promising research directions in the field of safe and trustworthy AI. First, while our current research focuses on image classification, there is potential to extend this approach to other vision and language tasks, such as object detection and machine translation. Second, future research could explore mutual learning between mentors and mentees, where mentors not only learn from the mentee's error patterns but also provide valuable feedback to help refine the mentee. Third, we can establish more rigorous evaluation criteria for mentors, broadening their predictive capabilities. For example, beyond predicting whether a mentee is likely to make errors, mentors could also forecast the specific types of errors a mentee may encounter. Fourth, this concept can be applied to investigate recognition errors in humans and primates, drawing parallels with AI models. Such analysis could provide insights into error pattern alignment between biological and artificial intelligent systems. Overall, our work lays the foundation for developing systems capable of anticipating the errors of others, offering practical value in high-stakes real-world applications.

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

## S1 DETAILED PERFORMANCE OF MENTORS ACROSS VARIOUS ERROR SOURCES

As mentioned in **Sec. 4.1**, the detailed results of mentors across various error sources for the CIFAR-10, CIFAR-100, ImageNet-1K datasets are shown in **Fig. S1**, **Fig. S2** and **Fig. S3** respectively.

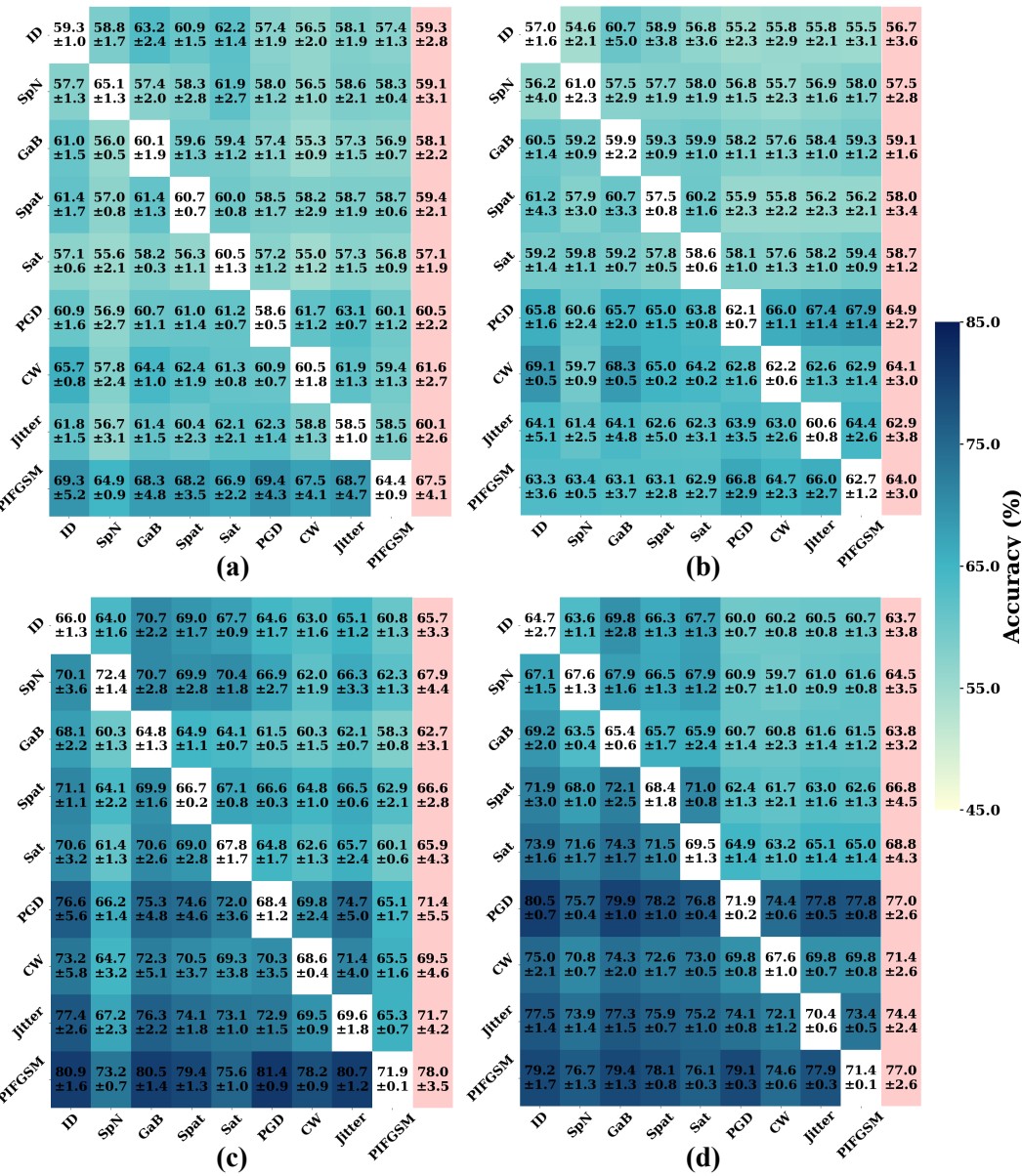

Figure S1: **Heatmaps showing the average performance of mentor models across various error sources for the CIFAR-10 dataset, presented in the format [mentee]-[mentor]: a) ResNet50-ResNet50, b) ViT-ResNet50, c) ResNet50-ViT, and d) ViT-ViT.** The heatmaps' row labels indicate the training error source for the mentor, while the column labels denote the testing error sources for the mentor. Results in each cell denote the average accuracy with the standard deviation over 3 runs. The pink-highlighted column displays the row-wise mean and standard deviation.

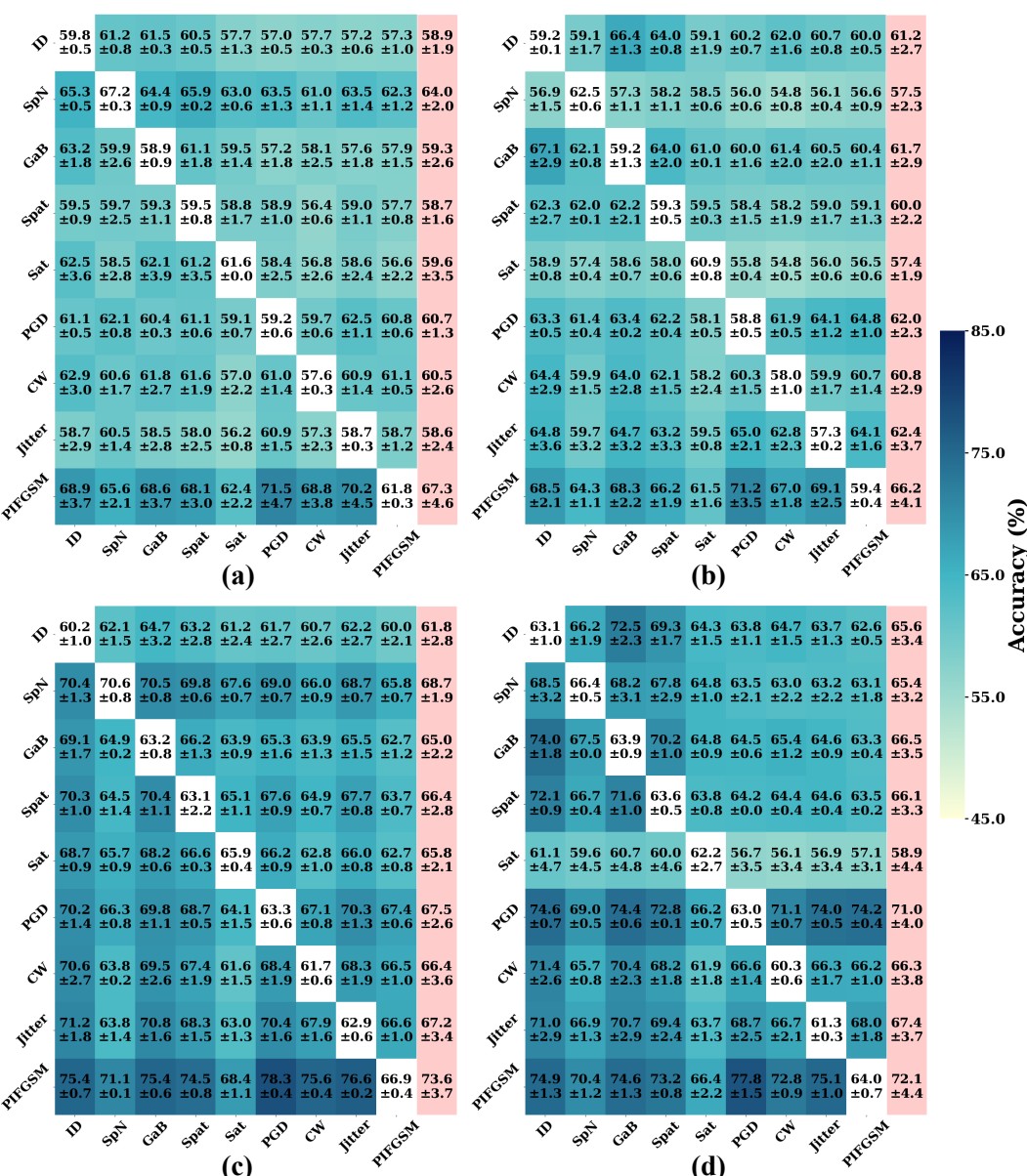

Figure S2: **Heatmaps showing the average performance of mentor models across various error sources for the CIFAR-100 dataset, presented in the format [mentee]-[mentor]: a) ResNet50-ResNet50, b) ViT-ResNet50, c) ResNet50-ViT, and d) ViT-ViT.** The heatmaps' row labels indicate the training error source for the mentor, while the column labels denote the testing error sources for the mentor. Results in each cell denote the average accuracy with the standard deviation over 3 runs. The pink-highlighted column displays the row-wise mean and standard deviation.

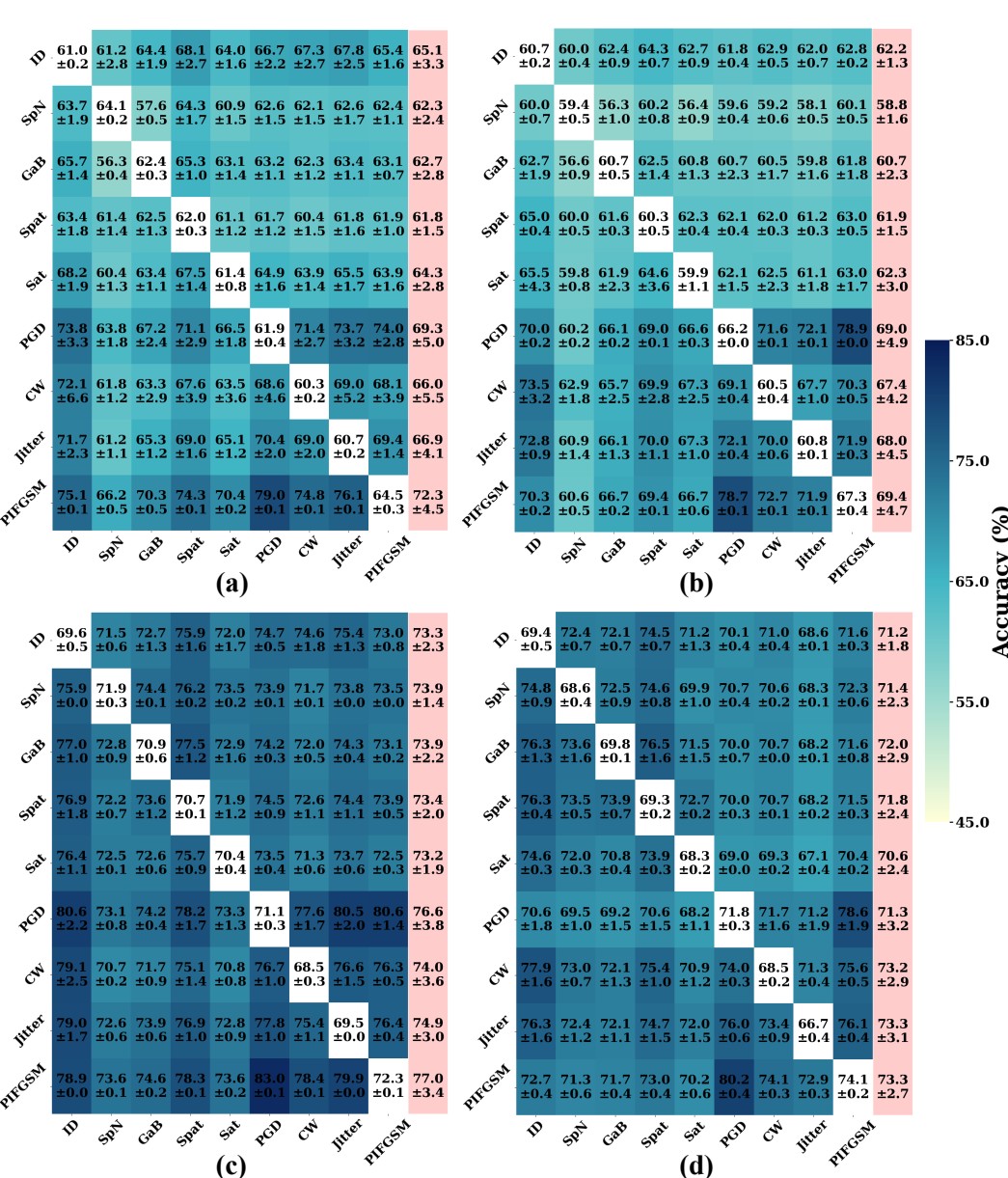

Figure S3: **Heatmaps showing the average performance of mentor models across various error sources for the ImageNet-1K dataset, presented in the format [mentee]-[mentor]: a) ResNet50-ResNet50, b) ViT-ResNet50, c) ResNet50-ViT, and d) ViT-ViT.** The heatmaps' row labels indicate the training error source for the mentor, while the column labels denote the testing error sources for the mentor. Results in each cell denote the average accuracy with the standard deviation over 3 runs. The pink-highlighted column displays the row-wise mean and standard deviation.

## S2   DETAILED PERFORMANCE OF MENTORS ACROSS MENTEE ARCHITECTURES

In **Fig. 5**, we show the generalization performance of mentors averaged over all three error types of mentees with various architectures. Here, we expand the results in the form of tables listing out all the individual accuracy for all the error sources on CIFAR-10, CIFAR-100 and ImageNet-1K datasets in **Tab. S1**, **Tab. S2**, and **Tab. S3** respectively.

| Mentor | | ResNet50 | | ViT | |
|---|---|---|---|---|---|
| Mentee | | ResNet50→ ViT | ViT→ ResNet50 | ResNet50→ ViT | ViT→ ResNet50 |
| ID | | 59.3±2.8→ 59.6±2.0 | 56.7±3.6→ 54.5±2.4 | 65.7±3.3→ 65.6±3.4 | 63.7±3.8→ 62.5±3.0 |
| OOD | SpN | 59.1±3.1→ 58.4±3.3 | 57.5±2.8→ 57.4±2.1 | 67.9±4.4→ 66.1±5.0 | 64.5±3.5→ 62.9±2.7 |
| | GaB | 58.1±2.2→ 58.2±2.0 | 59.1±1.6→ 57.3±1.8 | 62.7±3.1→ 62.4±2.8 | 63.8±3.2→ 62.6±2.8 |
| | Spat | 59.4±2.1→ 58.4±2.1 | 58.0±3.4→ 57.4±3.0 | 66.6±2.8→ 66.1±3.7 | 66.8±4.5→ 65.2±3.8 |
| | Sat | 57.1±1.9→ 57.7±1.3 | 58.7±1.2→ 56.7±1.8 | 65.9±4.3→ 66.4±3.2 | 68.8±4.3→ 67.7±4.3 |
| AA | PGD | 60.5±2.2→ 58.4±1.5 | 64.9±2.7→ 62.3±1.7 | 71.4±5.5→ 69.0±4.5 | 77.0±2.6→ 72.3±4.1 |
| | CW | 61.6±2.7→ 59.2±2.2 | 64.1±3.0→ 60.3±2.0 | 69.5±4.6→ 67.7±4.0 | 71.4±2.6→ 67.6±3.1 |
| | Jitter | 60.1±2.6→ 59.3±1.8 | 62.9±3.8→ 60.1±3.3 | 71.7±4.2→ 70.4±3.5 | 74.4±2.4→ 70.4±3.8 |
| | PIFGSM | 67.5±4.1→ 65.3±3.4 | 64.0±3.0→ 62.1±2.3 | 78.0±3.5→ 75.4±3.4 | 77.0±2.6→ 72.5±3.9 |
| Average | | 60.3±3.9→ 59.4±3.2 | 60.7±4.2→ 58.7±3.4 | 68.8±5.9→ 67.7±5.1 | 69.7±6.2→ 67.1±5.2 |

Table S1: **Detailed generalization performance of mentors across various mentee architectures on error sources from the CIFAR-10 dataset.** The mentee rows are formatted as [mentee A]→ [mentee B], as explained in **Fig. 5**. Results in each cell denote the average accuracy with the standard deviation over 3 runs.

| Mentor | | ResNet50 | | ViT | |
|---|---|---|---|---|---|
| Mentee | | ResNet50→ ViT | ViT→ ResNet50 | ResNet50→ ViT | ViT→ ResNet50 |
| ID | | 58.9±1.9→ 58.4±1.8 | 61.2±2.7→ 60.1±2.1 | 61.8±2.8→ 61.4±2.9 | 65.6±3.4→ 63.8±2.0 |
| OOD | SpN | 64.0±2.0→ 61.8±3.5 | 57.5±2.3→ 57.9±2.9 | 68.7±1.9→ 65.1± 3.2 | 65.4±3.2→ 64.6±2.6 |
| | GaB | 59.3±2.6→ 58.6±2.0 | 61.7±2.9→ 61.0±1.9 | 65.0±2.2→ 63.8±2.5 | 66.5±3.5→ 64.6±1.9 |
| | Spat | 58.7±1.6→ 57.7±1.9 | 60.0±2.2→ 59.5±2.0 | 66.4±2.8→ 64.9±2.7 | 66.1±3.3→ 64.0±2.0 |
| | Sat | 59.6±3.5→ 58.8±3.4 | 57.4±1.9→ 57.4±2.6 | 65.8±2.1→ 64.4±3.0 | 58.9±4.4→ 58.4±4.4 |
| AA | PGD | 60.7±1.3→ 59.4±1.1 | 62.0±2.3→ 60.6±1.3 | 67.5±2.6→ 65.5±2.0 | 71.0±4.0→ 67.2±2.1 |
| | CW | 60.5±2.6→ 59.2±1.8 | 60.8±2.9→ 58.9±2.1 | 66.4±3.6→ 64.5±2.1 | 66.3±3.8→ 63.6±1.8 |
| | Jitter | 58.6±2.4→ 57.8±2.0 | 62.4±3.7→ 60.5±2.4 | 67.2±3.4→ 65.2±2.2 | 67.4±3.7→ 65.1±1.9 |
| | PIFGSM | 67.3±4.6→ 64.6±2.5 | 66.2±4.1→ 64.3±2.2 | 73.6±3.7→ 69.1±1.7 | 72.1±4.4→ 68.5±2.3 |
| Average | | 60.8±3.9→ 59.6±3.2 | 61.0±3.8→ 60.0±2.9 | 66.9±4.1→ 64.9±3.2 | 66.6±5.2→ 64.4±3.6 |

Table S2: **Detailed generalization performance of mentors across various mentee architectures on error sources from the CIFAR-100 dataset.** The mentee rows are formatted as [menteeA]→ [menteeB], as explained in **Fig. 5**. Results in each cell denote the average accuracy with the standard deviation over 3 runs.

| Mentor | | ResNet50 | | ViT | |
|---|---|---|---|---|---|
| Mentee | | ResNet50→ ViT | V→ ResNet50 | ResNet50→ ViT | ViT→ ResNet50 |
| ID | | 65.1±3.3→ 63.9±2.7 | 62.2±1.3→ 62.5±1.6 | 73.3±2.3→ 71.0±2.4 | 71.2±1.8→ 71.5±1.4 |
| OOD | SpN | 62.3±2.4→ 62.0±3.7 | 58.8±1.6→ 58.4±2.2 | 73.9±1.4→ 71.1±2.2 | 71.4±2.3→ 71.2±1.3 |
| | GaB | 62.7±2.8→ 62.9±3.8 | 60.7±2.3→ 60.5±2.7 | 73.9±2.2→ 71.8±3.0 | 72.0±2.9→ 71.5±1.7 |
| | Spat | 61.8±1.5→ 61.1±1.7 | 61.9±1.5→ 61.7±1.4 | 73.4±2.0→ 71.4±2.3 | 71.8±2.4→ 71.7±1.4 |
| | Sat | 64.3±2.8→ 63.9±3.5 | 62.3±3.0→ 62.1±2.7 | 73.2±1.9→ 71.2±2.6 | 70.6±2.4→ 70.3±1.2 |
| AA | PGD | 69.3±5.0→ 66.8±2.6 | 69.0±4.9→ 67.1±2.7 | 76.6±3.8→ 72.6±2.0 | 71.3±3.2→ 70.3±1.6 |
| | CW | 66.0±5.5→ 64.3±3.5 | 67.4±4.2→ 65.8±2.3 | 74.0±3.6→ 71.1±1.9 | 73.2±2.9→ 71.7±1.4 |
| | Jitter | 66.9±4.1→ 65.3±2.5 | 68.0±4.5→ 66.1±2.6 | 74.9±3.0→ 72.2±2.0 | 73.3±3.1→ 71.3±1.3 |
| | PIFGSM | 72.3±4.5→ 69.3±2.2 | 69.4±4.7→ 68.0±2.9 | 77.0±3.4→ 72.9±1.7 | 73.3±2.7→ 72.1±1.4 |
| Average | | 65.6±5.0→ 64.4±3.8 | 64.4±5.1→ 63.6±3.9 | 74.4±3.0→ 71.7±2.4 | 72.0±2.8→ 71.3±1.5 |

Table S3: **Detailed generalization performance of mentors across various mentee architectures on error sources from the ImageNet-1K dataset.** The mentee rows are formatted as [menteeA]→ [menteeB], as explained in **Fig. 5**. Results in each cell denote the average accuracy with the standard deviation over 3 runs.

## S3 DETAILED RESULTS OF THE ABLATION STUDY ON THE LOSS COMPONENTS IN SUPERMENTOR

Extending the results shown in **Tab. 2**, we now include their standard deviations after 3 runs as presented in **Tab. S4**.

| | $L_d$ | $L_a$ | ID | OOD | | | | AA | | | | Average |
|---|---|---|---|---|---|---|---|---|---|---|---|---|
| | | | | SpN | GaB | Spat | Sat | PGD | CW | Jitter | PIFGSM | |
| C10 | ✗ | ✗ | 57.5± 1.2 | 61.0± 0.8 | 56.1± 1.1 | 58.6± 0.6 | 54.3± 1.5 | 58.6± 0.5 | 59.1± 1.2 | 58.5± 1.0 | 59.6± 0.6 | 58.2± 2.0 |
| | ✗ | ✓ | 80.0± 1.8 | 73.7± 0.6 | 79.2± 2.0 | 77.9± 0.8 | 74.3± 1.9 | 80.5± 0.8 | 76.5± 0.8 | 79.7± 0.7 | 71.2± 0.4 | 77.0± 3.2 |
| | ours | | 80.9± 1.6 | 73.2± 0.7 | 80.5± 1.4 | 79.4± 1.3 | 75.6± 1.0 | 81.4± 0.9 | 78.2± 0.9 | 80.7± 1.2 | 71.9± 0.1 | **78.0± 3.5** |
| C100 | ✗ | ✗ | 56.8± 1.2 | 59.5± 0.8 | 56.6± 1.1 | 57.8± 1.2 | 53.7± 1.3 | 57.7± 1.9 | 57.3± 1.9 | 57.3± 1.7 | 57.1± 0.5 | 57.1± 1.8 |
| | ✗ | ✓ | 75.0± 0.7 | 70.9± 0.3 | 74.8± 0.7 | 74.1± 0.3 | 68.1± 0.8 | 78.1± 0.6 | 75.2± 0.6 | 76.2± 0.5 | 66.5± 1.0 | 73.2± 3.7 |
| | ours | | 75.4± 0.7 | 71.1± 0.1 | 75.4± 0.6 | 74.5± 0.8 | 68.4± 1.1 | 78.3± 0.4 | 75.6± 0.4 | 76.6± 0.2 | 66.9± 0.4 | **73.6± 3.7** |
| IN | ✗ | ✗ | 73.0± 4.2 | 70.1± 2.8 | 69.6± 3.1 | 72.8± 3.7 | 68.5± 3.4 | 75.8± 5.2 | 72.5± 4.2 | 73.6± 4.6 | 70.7± 0.5 | 71.9± 3.8 |
| | ✗ | ✓ | 78.7± 0.1 | 73.1± 0.2 | 73.6± 0.5 | 78.0± 0.1 | 73.2± 0.3 | 83.0± 0.1 | 78.4± 0.1 | 79.9± 0.1 | 72.2± 0.2 | 76.7± 3.6 |
| | ours | | 78.9± 0.0 | 73.6± 0.1 | 74.6± 0.2 | 78.3± 0.1 | 73.6± 0.2 | 83.0± 0.1 | 78.4± 0.1 | 79.9± 0.0 | 72.3± 0.1 | **77.0± 3.4** |

Table S4: **Detailed results of the ablation study on the loss components in SuperMentor.** $L_d$ denotes the distillation loss (see **Sec. 3.1**) and $L_a$ represents the alignment loss between the mentor's and mentee's predicted class labels. SuperMentor is evaluated on a ResNet50 mentee. Results in each cell denote the average accuracy with the standard deviation over 3 runs. The performance of SuperMentor is highlighted in grey.

