# OpenReview forum: "Unveiling AI's Blind Spots: An Oracle for In-Domain, Out-of-Domain, and Adversarial Errors"
_ICLR.cc/2025/Conference — Submitted to ICLR 2025_

### Official Review · Reviewer_nkCR · 2024-10-31

**Soundness:** 3
**Presentation:** 4
**Contribution:** 4
**Rating:** 8
**Confidence:** 4

**Summary:**

This paper explores the utilization of a “mentor” model to predict the errors of a “mentee” model in a supervised image classification task. The strategy implies training the mentor on the frozen mentee by inputing the image, the mentee’s logits, and the correctness of the mentee in order to extract insight about the mentee’s decision patterns. The mentee’s logits are introduced in a distillation loss to incorporate its behaviour, while the prediction “correctness” (mentee's correctness vs. metor's prediction of mentee correctness) is supervised by a logistic regression loss.
In the experiments two types of architectures are used, ResNet50 and ViT. The latter corresponds to the powerful backbone since it is a transformer architecture based on self-attention mechanisms. Authors differentiate between 3 types of errors: In Domain errors, Out-of-domain errors and Adversarial Attack errors. Experiments are carried on these 3 types of errors to explore which reveals the mentee’s learning patterns better. According to the authors, it is important to remember that the mentor is expected to detect errors, not the source of them.
The experiments are carried over 3 datasets: CIFAR10, CIFAR100 and ImageNet-1K. For ID errors they use 3 different subsets of the original ones; for OOD they use 4 different corruptions for each dataset; and for the AA errors they use 4 different white-box attacks for each dataset.
The results show interesting findings:

1. **AA errors** offer deeper insights into the mentee’s decision-making process compared to OOD and ID errors.
2. Mentors benefit from having more **complex architectures** than the mentees, particularly transformer-based models.
3. The mentor's performance **degrades with increasing perturbation** in AA examples.
4. Most notably, **mentors generalize across different mentee architectures**, allowing them to predict errors even when mentees change.

As a final contribution, authors propose an “oracle” mentor based on the findings from the experiments. This SuperMentor is trained on AA with small perturbation and its architecture correponds to ViT. An ablation study is performed on this mentor to determine the contribution of the distillation loss and the mentee’s logits on its performance. This SuperMentor is able to correctly separate most of the correct/incorrect points of different mentees over different datasets and sources of errors.

**Strengths:**

- The problem assessed in the paper is an important one. Even though there exist other approaches for selective classification like, for example, Learning to Defer and Rejection Learning, the presented framework allows working with pretrained models and uses convex losses, thus avoiding the use of surrogate losses.
- The experiments are numerous and well chosen. The use of different error sources helps to understand the mentor’s understanding of the mentee. The use of different architectures is also an important one and the conclusion over them is insightful.
- Experimental results are exhaustive, well explained and supported in the text.

**Weaknesses:**

- I miss a bit of mathematical formulation in Section 3. I would like to see the complete loss function somewhere (even though it can be easily understood in the text).

- Since the framework is intended to detect potential errors regardless of their domain, I miss some references on Selective Classification, Rejection Learning [1] and Learning to Defer [2].

- The paper would benefit from comparing and discussing supervised and unsupervised strategies to solve the issues dealt by their mentor system. See the next point.

- The authors try to distinguish and characterize three types of errors, but there is conflict in OOD errors and AA. For instance AA are not necessarily mutually exclusive with OOD, moreover trying to describe OOD errors with a set of perturbations on original images is not exhaustive or representative of OOD behaviour. In the literature on OOD detection there is justified skepticism towards using supervised examples of OOD, issue that is completely ignored across the paper and not mentioned as a possible limitation.

[1] Cortes, C., DeSalvo, G., & Mohri, M. (2016). Learning with rejection. In *Algorithmic Learning Theory: 27th International Conference, ALT 2016, Bari, Italy, October 19-21, 2016, Proceedings 27* (pp. 67-82). Springer International Publishing.
[2] Madras, D., Pitassi, T., & Zemel, R. (2018). Predict responsibly: improving fairness and accuracy by learning to defer. *Advances in neural information processing systems*, *31*.

**Questions:**

- Can you elaborate on how does your model respond in multi-class setting in which two classes could be equally likely but the model makes a mistake in choosing the correct label?
- Did authors consider what to do next once the mentor predicts an error? How could the information be used?
- Are there any external baselines to compare against? If so, why aren’t they included?
- How would you tackle the calibration of the mentor’s predictions?
- Considering that the mentor is trained on examples of OOD or AA, there is a chance that the mentor learns to detect these kinds of data but nothing actually related to the mentee. This is similar to the faithfulness issue in explainability, where we might learn plausible explanations that have nothing to do with the method to explain. Could you comment on this point?

---

> ### Author Response · Authors · 2024-11-23
> **Response to the Reviewer nkCR (Part 1)**
>
> **[nkCR.1-Weakness] I miss a bit of mathematical formulation in Section 3. I would like to see the complete loss function somewhere (even though it can be easily understood in the text).**
>
> The distillation loss formula $L_d$ is:
> $L_d = KL(\sigma(\frac{z_E}{T})||\sigma(\frac{z_R}{T}))$, where $KL(\cdot||\cdot)$ represents Kullback-Leibler divergence, $\sigma(\cdot)$ is the softmax function and $T$ is the temperature.
>
> The logistic regression loss $L_r$ is:
>
> $ L_r = - \left[ c_E \log(z_p) + (1 - c_E) \log(1 - z_p) \right]$, where $c_E$ is the ground truth correctness label of a mentee and $z_p$ represents the mentor’s predicted probability that the mentee’s prediction is correct.
>
> We will add these formulas in the final version.
>
> **[nkCR.2-Weakness]  Since the framework is intended to detect potential errors regardless of their domain, I miss some references on Selective Classification, Rejection Learning [1] and Learning to Defer [2].**
>
> Thank you for highlighting the amazing works [d-f]. They share a similar focus with our work, as all of them aim to address the topic on trustworthy AI. However, there are several key differences between our work and these methods, which are outlined below:
>
> **Selective Prediction & Rejection Learning:**
>
> Selective Prediction [f] and Rejection Learning [d] aim to minimize a model's error rate by choosing to reject unreliable predictions. Our work differs from theirs in three key aspects:
>
> 1) The selection function in [f] is not implemented as a DNN, whereas the mentors in our work are deep neural networks (DNNs). This distinction allows our mentor to effectively learn and adapt to the mentee's complex learning patterns. In contrast, the approach in [f] may struggle to capture and leverage such intricate patterns for error prediction.
>
> 2) The selection function in [f] and the rejection function in [d] are trained jointly with the model, whereas in our work, the mentor and mentee are trained independently. This independence is a significant advantage, particularly when training the mentee is resource-intensive (e.g., requiring extensive training data, prolonged training time, or high memory usage) or when the mentee's training data is inaccessible. In such cases, the methods in [f] and [d] may face substantial challenges or even fail, whereas our mentor operates without these constraints.
>
> 3) The selection and rejection functions are designed to identify confident predictions or discard unconfident ones, respectively. While there is often a positive correlation between a mentee's confidence and its error rates, these are distinct from directly predicting the correctness of the model's outputs. This distinction is evident in the results of the three competitive baselines discussed in our general response to all reviewers “**Common.1: Performance comparison of mentor baselines and our SuperMentor**”. For instance, using the entropy of the mentee's logits as an error indicator achieves above-chance performance; however, its error prediction accuracy remains lower than that of our SuperMentor.
>
> **Learning to Defer:**
>
> Learning to Defer [e] extends rejection learning by incorporating the influence of other agents in the decision-making process, such as human experts or more specialized models. Our work differs from theirs in two key aspects:
>
> 1) The model in [e] jointly outputs predictive probabilities for downstream tasks and binary deferral decisions within a single network. In contrast, our framework separates the mentee and mentor into distinct networks. This separation provides greater flexibility for the mentor, as it can predict error patterns for multiple mentees without requiring retraining for each individual mentee. Conversely, the approach in [e] lacks this flexibility, as it combines both outputs in a single model, necessitating retraining and the addition of a new branch to produce binary deferral decisions on the new mentees.
>
> 2) The decision maker in [e] may rely on human experts who are not specifically trained to understand the behaviors of the model producing binary deferral decisions. As a result, the decision-making process may not align well with the true error patterns of the model. Additionally, involving humans in the loop is resource-intensive and costly. In contrast, our mentor is an AI model specifically designed to learn and understand the behaviors of mentees. This enables it to make error predictions based on these behaviors, resulting in more robust, efficient, and informed decision-making.
>
> We will cite these works and add these discussions in the final version.

---

> ### Author Response · Authors · 2024-11-23
> **Response to the Reviewer nkCR (Part 2)**
>
> **[nkCR.3-Weakness] The paper would benefit from comparing and discussing supervised and unsupervised strategies to solve the issues dealt by their mentor system. See the next point.**
>
> **[nkCR.4-Weakness] The authors try to distinguish and characterize three types of errors, but there is conflict in OOD errors and AA. For instance AA are not necessarily mutually exclusive with OOD, moreover trying to describe OOD errors with a set of perturbations on original images is not exhaustive or representative of OOD behaviour. In the literature on OOD detection there is justified skepticism towards using supervised examples of OOD, issue that is completely ignored across the paper and not mentioned as a possible limitation.**
>
> We do agree with the reviewer that AA is not necessarily mutually exclusive with OOD. To validate this point, we now added an experiment by training a Vision Transformer (ViT)-based mentor model on the correctness of a mixture of ID, OOD, and AA data for the ResNet-based mentee model from the CIFAR-10 dataset. The OOD data was corrupted with speckle noise, and the AA data was generated using PIFGSM. Compared to training the mentor solely on AA data which achieved an average accuracy of 78.0%, including ID and OOD data improved the average accuracy slightly to 79.1%. This minor improvement suggests a significant overlap of mentee error patterns among ID, OOD, and AA data. In other words, training mentors on OOD or ID data do not necessarily benefit mentors. Therefore, considering the associated computational costs, it is recommended to train mentors exclusively on AA data in practical applications. This finding underscores the significance of our study by providing critical insights into the optimal training practices for mentor models.
>
> Moreover, we agree with the reviewer on the skepticism of the supervised training OOD samples in the literature. However, we respectfully disagree with the reviewer that our SuperMentor has such limitations.
>
> First, our SuperMentor is trained solely on one specific type of AA images. It has not been trained on any OOD images. Surprisingly, our SuperMentor can still generalize to predict errors from OOD samples.
>
> Second, we also agree with the reviewer that describing OOD errors with a set of perturbations on original images is not exhaustive or representative of OOD behaviour. To rigorously assess the generailzation ability of our SuperMentor, we have added experiments to evaluate the SuperMentor on more OOD domains in ImageNet dataset, as described in the general comment to all Reviewers “**Common.2: SuperMentor performance on additional OOD domains on ImageNet dataset**”. Despite zero training on these OOD domains, it is remarkable that our SuperMentor achieves above-chance performance.
>
> We will add these new experimental results and provide discussions about limitations on supervised training of OOD examples in the final version.

---

> ### Author Response · Authors · 2024-11-23
> **Response to the Reviewer nkCR (Part 3)**
>
> **[nkCR.5-Questions] Can you elaborate on how does your model respond in multi-class setting in which two classes could be equally likely but the model makes a mistake in choosing the correct label?**
>
> We would like to point out that all the reported accuracies of our SuperMentor in the main text are calculated by averaging the mentor's accuracies on the samples that the mentee classified correctly and those that the mentee classified incorrectly. In other words, we have taken into account the long-tailed distribution of the binary classification task. For example, if the mentee correctly classified 10 samples and incorrectly classified 90 samples, and the mentor accurately predicts the correctness of 8 out of the 10 correctly classified samples and 9 out of the 90 incorrectly classified samples, the mentor's accuracy would be calculated as (8/10+9/90)/2 = 45%. We will clarify this point in the final version.
>
> As the reviewer suggested here, we now included an additional experiment evaluating the mentor's performance on two classes from the CIFAR-10 dataset, specifically "airplane" and "dog," referred to as CIFAR-2. In this experiment, we trained a Vision Transformer (ViT)-based mentor model alongside a ResNet-based mentee model. The results presented in Tab. R5 demonstrates that the mentor accurately determines the correctness of the mentee’s predictions with an accuracy of up to 80.9 %. The highest accuracy is achieved by training the mentor on the mentee's adversarial images generated by PIFGSM, which aligns with the SuperMentor configurations in our study.
>
> Although this binary classification problem is relatively straightforward, leading to fewer mistakes from the mentee, the OOD and AA scenarios remain challenging. The mentor's performance significantly exceeds the 50% chance level, demonstrating its promising error predictive capabilities.
>
> We will add these new experimental results and discussions in the final version.
>
> **Table R5: The average accuracy of our SuperMentor trained on various error sources for the CIFAR-2 datasets. The mentor and mentee are ViT and ResNet50 respectively. The CIFAR-2 dataset stands for only using two classes (“airplane” and “dog”) in the CIFAR-10 dataset for classification. Results denote the average accuracy over 3 runs.**
> |              |      ID     |     SpN    |     GaB     |     Spat    |     Sat     |     PGD     |      CW     |    Jitter   |    PIFGSM   |
> |--------------|:-----------:|:----------:|:-----------:|:-----------:|:-----------:|:-----------:|:-----------:|:-----------:|:-----------:|
> |    CIFAR-2    | 59.9 | 62.0 | 66.4 | 66.2 | 69.0 | 71.9 | 74.8  | 64.0 | 80.9  |
>
>
> **[nkCR.6-Questions] Did authors consider what to do next once the mentor predicts an error? How could the information be used?**
>
> That is a very good point!
>
> This exciting new problem of using AI to predict the errors of another AI opens numerous research directions. For instance, one potential direction is employing GradCAM [h] to visualize the mentees' error patterns detected by our mentor at the pixel level. The mentee can use these GradCAM-generated images to enhance their learning. Another avenue could involve mutual learning between mentors and mentees, where mentors not only learn from the mentees' errors but also provide constructive feedback to refine the mentees' performance by acting as a discriminator to the mentee. We will add these discussions in the final version.
>
> **[nkCR.7-Questions] Are there any external baselines to compare against? If so, why aren’t they included?**
>
> To the best of our knowledge, our study is the first to use one AI model to predict the correctness of another AI model’s predictions.
> Due to the lack of existing AI model baselines, as asked by the reviewer, we now introduced three baseline approaches based on the mentee’s confidence, the entropy of the mentee’s logits, and the L2 distance between the mentee’s features and the class feature centroids, respectively. The details of experimental results are described in the general comment to all Reviewers “**Common.1:  Performance comparison of mentor baselines and our SuperMentor**”.
> We will add these new experiments in the final version.
>
>
> **[nkCR.8-Questions] How would you tackle the calibration of the mentor’s predictions?**
>
> Our current work prioritizes enhancing the mentee's error prediction capabilities over calibrating the mentor’s predictions. However, we agree that calibrating the mentor’s predictions is a crucial direction for future research to prevent the mentor from becoming overconfident or underconfident. We appreciate the reviewer bringing this point to our attention.
>
> We will add these discussions in the final version.

---

> ### Author Response · Authors · 2024-11-23
> **Response to the Reviewer nkCR (Part 4)**
>
> **[nkCR.9-Questions] Considering that the mentor is trained on examples of OOD or AA, there is a chance that the mentor learns to detect these kinds of data but nothing actually related to the mentee. This is similar to the faithfulness issue in explainability, where we might learn plausible explanations that have nothing to do with the method to explain. Could you comment on this point?**
>
> This is a brilliant point!
>
> On one hand, the reviewer points out the potential issue of shared vulnerabilities between the mentor and mentee models, which might lead to faithfulness issues in explainability. This is a critical aspect to consider when designing systems that rely on one deep neural network (DNN) to evaluate another. However, we respectfully disagree with the reviewer that both mentees and mentors share the same vulnerability. On the other hand, we argue that even though these challenges on the faithfulness issue in explainability may exist, they will not hinder the deployment of our framework for error prediction in real-world practice. Next, we elaborate on these two aspects below.
>
> First, in our approach, we intentionally design the mentor model to specialize in detecting and quantifying vulnerabilities of the mentee, rather than solving the primary task of the mentee. This difference in objective between mentors and mentees helps mitigate the risk of overlapping weaknesses. For example, while the mentee might struggle with generalization in a classification task, the mentor focuses solely on identifying patterns indicative of vulnerability, which is a different domain.
>
> Second, the mentor model is designed with robustness in mind. Despite zero training on many error sources in the main text as well as the newly added challenging OOD domains in the rebuttal (see “**Common.2: SuperMentor performance on additional OOD domains on ImageNet dataset**”), our mentor achieves above-chance performance in detecting errors of the mentees. Moreover, we introduce three competitive baselines and our mentor surpasses the performance of these mentor baselines. (see “**Common.1: Performance comparison of mentor baselines and our SuperMentor**”). All this evidence suggests that our mentors capture the intricate error patterns of the mentees. Otherwise, the mentors will fail to generalize to multiple OOD domains.
>
> Third, the mentor model is designed to be model-agnostic in its evaluations, meaning that it operates independently of the specific architecture or training details of the mentee. This reduces the risk of the mentor inheriting the mentee's specific weaknesses.
>
> Fourth, interestingly, if the mentor model does exhibit similar vulnerabilities, this can serve as an insightful diagnostic tool. Identifying such shared vulnerabilities can highlight systemic issues in the broader DNN training paradigm, offering opportunities to address them at a foundational level. Our framework provides a unique opportunity to improve our understanding of both individual and systemic vulnerabilities in DNNs, leading to more robust AI systems overall. One possible direction is to use GradCAM [h] to visualize the error patterns detected by our mentor at the pixel levels.
>
> Fifth, even though these challenges on the faithfulness issue in explainability may exist, they will not hinder the deployment of our framework for error prediction in real-world practice. As AI models become more integrated into our daily lives, ensuring their correctness is of paramount importance, especially in high-stakes fields such as the medical and finance fields. To demonstrate the usefulness of our mentor model in real-world applications, we expanded our experiments to a medical image classification task, as described in the general comment to all Reviewers “**Common.3: SuperMentor in real-world practice, such as NCTCRCHE100K dataset**”.
>
> Sixth, we argue that our proposed framework of using one DNN to automatically discover the error patterns of the other DNN provides a unique opportunity to improve our understanding of both individual and systemic vulnerabilities in DNNs, leading to more robust AI systems overall. For example, AlphaFold [i] has revolutionized protein structure prediction, and DNNs are successfully employed in detecting diabetic retinopathy from retinal images [j] —identifying patterns that even doctors had not previously recognized. One advantage of these DNN models is that we no longer need to manually design hand-crafted features for downstream tasks. Instead, DNNs automatically discover these intricate patterns for us.
>
> We will add these experiments and discussions in the final version.

---

> ### Comment · Reviewer_nkCR · 2024-11-28
> **Final answer and appreciation**
>
> Thank you for the deep and detailed explanation of all the pointed concerns. I am happy to see that this review created a great discussion over this interesting paper.
> I appreciate you took time to write the mathematical formulation, which I already had clear but I think it should appear in the paper. By the way, as far as I know, the distillation loss is normally scaled by T² to contrast the 1/T² scaling of the gradient (since your are using T=1 this makes no difference, I know). I also appreciate your arguments for separating frameworks like Rejection Learning, Learning to Defer and Selective Classification from your approach, I agree on all of them. It is also impressing the amount of extra effort to clarify the questions answered in the Common responses.
>
> Despite all of that I will keep my score as initially because a new concern has risen based on your answers. When accuracy is measured the way you do it is natural that question like that of Reviewer pVLx appear (the question regarding an imbalanced task). Following your example, if your mentor classifies all 10 correctly classified samples as they are but none of the 90 misclassifications, the accuracy will be (10/10+0/90)/2=50%. However, globally, it correctly labeled less than the 45% example you gave, 10<17. In terms of trustworthiness this metrics does not faithfully represent a reliable mentor. Moreover, (0/10+90/90)/2=50% too but in terms of safety this is a safer alternative because we would be detecting 100% of errors.
>
> Thank you again for your rebuttal.

---

> > ### Author Response · Authors · 2024-11-30
> > **Round 2 - Reviewer nkCR**
> >
> > We thank the reviewer for raising the question about our evaluation metric! Indeed, every metric has its strengths and weaknesses. We will provide results with more evaluation metrics, such as Precision, Recall (TPR), and Specificity (TNR)  in the final version. If the reviewer has more suggestions, please feel free to let us know.

---

### Official Review · Reviewer_K1t1 · 2024-11-02

**Soundness:** 3
**Presentation:** 3
**Contribution:** 3
**Rating:** 5
**Confidence:** 4

**Summary:**

The paper proposes using mentor model to predict the error of AI models (mentee models). It mainly considers image classification task with two architectures, ResNet50 and ViT. The basic idea of the paper is similar to membership inference attack(MIA), which treats the mentor as a binary classifier. The paper conducts decent empirical studies on different setup of data distribution, including in-domain, out-of-domain, and adversarial examples. The paper makes several interesting conclusions, including: 1. adversarial examples with small perturbations are most beneficial to mentor model training. 2. Mentor models can generalize across different mentee models.

**Strengths:**

1. The formulation of the problem is interesting. The paper proposes using an AI model to predict the error of another, which can open new possibility for understanding DNNs.
2. The experiments are decent and comprehensive. The paper conducts experiments on multiple setup of model architecture and data distribution. It also includes experiments like generalizability of the model. In particular, the incorporation of AA data provides interesting insights.
3. The paper is generally well written and easy to follow.

**Weaknesses:**

1. The paper lacks a comprehensive assessment of relationship to previous works. Monitoring errors of AI model seem crucial, but how previous approach attempts to address the problem is not clear (although it is presented in related work). The paper also does not compare its performance to previous methods, nor does it explicitly claim that the it is formalizing a new problem, which makes understanding the position and results of the paper in the area relatively difficult.

2. This is like my main concern -- I am very willing to discuss this further with the authors. It is not clear to me what the possible applications of the method is/what possible insights we could draw from the mentor.  For example, is it possible to give insights on interpretation of the mentee model based on the mentor model?

3. (minor) The accuracy of the mentor model seems not high "enough" -- although it outperforms random guessing by a large amount, it seems not high enough to safely draw more insights independently from it. Again it is related to first point as the position of the paper is unclear.

**Questions:**

1. See Weaknesses, especially weakness 2.

2. In the results section, why is it sometimes possible that mentor model trained on ID data can outperform that trained on OOD data?

3. Is it possible that a mixture of ID, OOD and AA data can achieve better performance? AA data is like outliers for the mentee model, which, in turn, may lack some information of the in-domain distribution that the mentee model holds.

---

> ### Author Response · Authors · 2024-11-23
> **Response to the Reviewer K1t1 (Part 1)**
>
> **[K1t1.1-Weakness] The paper lacks a comprehensive assessment of relationship to previous works. Monitoring errors of AI model seem crucial, but how previous approach attempts to address the problem is not clear (although it is presented in related work). The paper also does not compare its performance to previous methods, nor does it explicitly claim that the it is formalizing a new problem, which makes understanding the position and results of the paper in the area relatively difficult.**
>
>
> Thanks for pointing this out! As suggested by the reviewer, we will add the claim that our study is indeed the first to use one AI model to predict the correctness of another AI model’s predictions, to our best knowledge.
> Due to the lack of existing AI model baselines to tackle this new problem, as suggested by the reviewer, we now introduce three baseline approaches based on the mentee’s confidence, the entropy of the mentee’s logits, and the L2 distance between the mentee’s features and the class feature centroids, respectively. The details of experimental results are described in the general comment to all Reviewers “**Common.1: Performance comparison of mentor baselines and our SuperMentor**”.
>
>
> We will add these discussions to the related work and add these new experimental results in the final version.
>
> **[K1t1.2-Weakness] This is like my main concern -- I am very willing to discuss this further with the authors. It is not clear to me what the possible applications of the method is/what possible insights we could draw from the mentor. For example, is it possible to give insights on interpretation of the mentee model based on the mentor model?**
>
>
> As AI models become more integrated into our daily lives, ensuring their correctness is of paramount importance, especially in high-stakes fields such as the medical and finance fields. First, as suggested by the reviewer, to demonstrate the usefulness of our mentor model in real-world applications, we expanded our experiments to a medical image classification task, as described in the general comment to all Reviewers “**Common.3: SuperMentor in real-world practice, such as NCTCRCHE100K dataset**”. We will add these experiments in the final version.
>
> Second, from our work, the main insights from the mentor are: 1)  Training mentors with adversarial attack errors from the mentee has the most significant impact on improving the mentor’s error prediction accuracy compared to ID and OOD errors. 2) Transformer-based mentor models outperform other architectures in accurately predicting errors. 3) Training mentors with images with small perturbations can improve error prediction accuracy. 4) A mentor trained to learn error patterns from one mentee can successfully generalize its error predictions to another mentee.
>
> Third, we also provide the loss landscape analysis of the mentee in Sec. 4.1 and Sec. 4.2. These insights on the mentees are obtained from the observations of the mentors’ behaviors. Summaries of key insights are: 1) Since adversarial images are positioned closer to the mentee’s decision boundary, training with these images allows more accurate predictions of the mentee’s errors and provides a deeper understanding of the loss landscape of the mentee. 2) Self-attention mechanism in ViT captures more complex relationships among data samples than convolution layers in ResNet50, particularly in modelling irregular, rugged landscapes with sharp peaks and valleys.
>
> We agree with the reviewer that there are definitely more analyses that we can perform to probe the error patterns of the mentees! We would be happy to connect with the reviewer and discuss other possible analyses we can perform! This exciting new problem of using AI to predict the errors of another AI opens numerous research directions. For instance, one potential direction is employing GradCAM [h] to visualize the mentees' error patterns detected by our mentor at the pixel level. Another avenue could involve mutual learning between mentors and mentees, where mentors not only learn from the mentees' errors but also provide constructive feedback to refine the mentees' performance. We will add these discussions in the final version.

---

> > ### Author Response · Authors · 2024-11-26
> >
> > Dear Reviewer,
> >
> > We have provided responses to all your questions and would greatly appreciate it if you could review them to see whether we have adequately addressed your concerns, particularly regarding **[K1t1.2-Weakness]**.
> >
> > Additionally, in your review, you mentioned, "I am very willing to discuss this further with the authors." If you have any further advice on analyses we could perform to gain deeper insights into the mentees, please let us know.
> >
> > Thank you for your time and feedback!

---

> ### Author Response · Authors · 2024-11-23
> **Response to the Reviewer K1t1 (Part 2)**
>
> **[K1t1.3-Weakness:(minor)] The accuracy of the mentor model seems not high "enough" -- although it outperforms random guessing by a large amount, it seems not high enough to safely draw more insights independently from it. Again it is related to first point as the position of the paper is unclear.**
>
> Thanks for pointing this out! As suggested by the reviewer, we will add the claim that our study is indeed the first to use one AI model to predict the correctness of another AI model’s predictions, to our best knowledge.
> Due to the lack of existing AI model baselines to tackle this new problem, as suggested by the reviewer, we now introduce three baseline approaches based on the mentee’s confidence, the entropy of the mentee’s logits, and the L2 distance between the mentee’s features and the class feature centroids, respectively. The details of experimental results are described in the general comment to all Reviewers “**Common.1: Performance comparison of mentor baselines and our SuperMentor**”. Results suggest that our mentor also outperforms these three baselines, in addition to the chance performance.
>
> We agree with the reviewer that there is considerable room to improve the mentor's error prediction capabilities. In this work, we take an initial step by proposing the SuperMentor model. Despite being a relatively simple ViT architecture, it is still quite remarkable to achieve above-chance error prediction performance. We hope our work inspires researchers to explore more sophisticated AI mentors, incorporating advanced architectures or loss functions, to further enhance the ability to predict the correctness of another AI model’s outputs.
>
> We will add these discussions and new experimental results in the final version.
>
> **[K1t1.4-Question] See Weaknesses, especially weakness 2.**
>
> See our replies to **[K1t1.2-Weakness]**.
>
> **[K1t1.5-Question] In the results section, why is it sometimes possible that mentor model trained on ID data can outperform that trained on OOD data?**
>
> The effectiveness of training data in enhancing the mentor's ability to predict the mentee's errors largely depends on the proximity of these data points to the mentee's decision boundary. In some cases, ID data may be closer to the decision boundary than OOD data, making it more effective for error prediction. However, AA data is specifically designed to consistently lie near the mentee's decision boundary. This consistent closeness makes AA data particularly valuable for improving the mentor's error prediction ability, as it exposes the mentor to scenarios where the mentee is most prone to errors. This example demonstrates how the mentor's error predictions can provide insights into the landscapes of the mentee's decision boundaries.
>
> We will add these discussions in the final version.
>
> **[K1t1.6-Question] Is it possible that a mixture of ID, OOD and AA data can achieve better performance? AA data is like outliers for the mentee model, which, in turn, may lack some information of the in-domain distribution that the mentee model holds.**
>
>
> That is a good point! As suggested by the reviewer, We conducted experiments by training a Vision Transformer (ViT)-based mentor model to predict the correctness of a ResNet-based mentee model’s performance using a mixture of ID, OOD, and AA data from the CIFAR-10 dataset. The OOD data was corrupted with speckle noise, and the AA data was generated using the PIFGSM attack. Compared to training the mentor solely on AA data which achieved an average accuracy of 78.0%, including ID and OOD data improved the average accuracy slightly to 79.1%.
>
> This minor improvement suggests a significant overlap of the mentee’s error patterns among ID, OOD, and AA data. In other words, training mentors on OOD or ID data do not necessarily benefit mentors greatly. Therefore, considering the associated computational costs, it is recommended to train mentors exclusively on AA data in practical applications. This finding underscores the significance of our study by providing critical insights into the optimal training practices for mentor models.
>
> Additionally, we discuss the loss landscape analysis of the mentee in Sections 4.1 and 4.2. Based on this analysis, AA data should not be considered outliers, as they lie close to the model's decision boundary. This proximity has been shown in our work to make AA data more effective than ID data for training the mentor, as it better highlights areas where the mentee is most likely to make errors.
>
> We will add these discussions and new experimental results in the final version.

---

> ### Comment · Reviewer_K1t1 · 2024-11-26
>
> Thanks for the authors' detailed response and the efforts they put on addressing my concerns. Most of my concerns are addressed, but I partially agree with Reviewer pVLx on the concern of risk introduced by the possible vulnerability of the mentor model, which I believe may significantly downgrade the trustworthiness of the insights we draw for the mentor model. Furthermore, from the results comparing SuperMentor with other baselines, it seems that SuperMentor does not lead in several tasks, although it is carefully designed and utilizing advanced strategy (DNN).
>
> I would keep my score for now based on the two remaining concerns. I would wait to see other reviewers' opinions, especially on the  vulnerability of the mentor model.

---

> ### Author Response · Authors · 2024-11-30
> **Round 2 - Reviewer K1t1 (Part 1)**
>
> We respectfully disagree with the Reviewer K1t1 for penalizing us on challenges of vulnerability common to all DNNs. We present the following arguments:
>
> 1.**Shared vulnerabilities across all DNNs do not stop us from deploying them in the real-world applications.**
>
> Vulnerability is a well-known challenge for all DNN-based systems, yet this has not hindered their transformative applications in the real world. For example, AlphaFold [i] revolutionized protein structure prediction, and DNNs are used to detect diabetic retinopathy [j], outperforming traditional diagnostic methods. Similarly, our work uses one AI to diagnose the other AI, which is a valid and impactful use case. To demonstrate real-world utility, we extended our experiments to a medical image classification task, as detailed in **Common.3**. Using the NCTCRCHE100K dataset for colorectal cancer diagnosis, our mentor model achieves high classification accuracy, showcasing its practical uses in high-stake applications.
>
> 2.**Our AI mentors are robust to error patterns of mentees even in OOD domains that the AI mentors have never been trained on.**
>
> We rigorously evaluate our mentor models across 13 OOD domains (8 in the main text and 5 newly added in the rebuttal). Despite no prior exposure to these domains, the mentor generalizes well, identifying mentee errors far above chance and surpassing all the competitive baselines. Please refer to **Common.2** and **Common.1** for detailed performance comparisons with all the competitive baselines.
>
> 3.**The vulnerabilities of the mentor AI do not overlap with those of the mentees. The mentor's vulnerabilities are less of a concern, as our ultimate goal is to address and eliminate the vulnerabilities of the mentees.**
>
> The vulnerabilities of mentors and mentees are distinct due to their non-overlapping objectives. While mentees may struggle with generalization in specific image classification tasks, mentors are designed to focus solely on the error pattern detection of the mentees. Furthermore, mentors operate independently of the mentee’s architecture or training details, reducing the risk of shared weaknesses. This distinction ensures that mentors effectively reduce the vulnerabilities of mentees that matter most in real-world applications.
>
> To support the above discussion, we use EigenCAM [o,p] to visualize the behavior of both our ViT SuperMentor and ViT mentee on example images, as depicted in Fig. R1 at the link [**HERE**](https://drive.google.com/file/d/1DA9yEQ4bf_4YdOjk9ds408Nwet6ERfX5/view?usp=sharing). As illustrated in Fig. R1, the mentor does not merely replicate the mentee’s learning patterns, as their activation maps for identical input images can differ significantly. For example, in the images shown in Fig. R1 (a), although the mentor and mentee exhibit very similar activation maps for ID images, their activations diverge greatly when the input image is corrupted by Gaussian Blur (GaB) or subjected to adversarial attacks using PIFGSM method. In the IN-GaB and IN-PIFGSM images, the mentee demonstrates vulnerability under both corruption and adversarial attacks, whereas our SuperMentor does not exhibit such vulnerabilities on these OOD and AA images, consistently focusing on the brambling object. This indicates the vulnerabilities of mentors and mentees are distinct due to their non-overlapping objectives.
>
> 4.**Our AI mentor serves as a valuable diagnostic tool for the AI mentees.**
>
> Beyond error detection, mentors serve as valuable diagnostic tools for AI mentees. Our framework improves the understanding of both individual and systemic vulnerabilities in DNN-based mentees, fostering more robust AI systems.
>  This is further illustrated by the EigenCAM [o,p] visualization in Figure R1 (link [**HERE**](https://drive.google.com/file/d/1DA9yEQ4bf_4YdOjk9ds408Nwet6ERfX5/view?usp=sharing)). For example, in Fig. R1(b), although the mentee correctly classifies the ID image, its activation map does not concentrate on the baseball but instead highlights background cues (spurious features). When the input image is corrupted by Gaussian Blur (GaB) or subjected to adversarial PIFGSM attacks, the mentee fails to make accurate predictions. In contrast, our SuperMentor effectively focuses on the stitches of the baseball—a key feature—regardless of whether the image is corrupted or adversarially attacked. The mentor highlights the stitches of the baseball as the key feature pattern that the mentee tends to overlook. This suggests that the mentor can identify vulnerabilities in DNN-based models, even though it shares the same ViT architecture as the mentee. Furthermore, this reveals significant potential for our mentor to help refine the mentee’s performance by providing valuable insights into the mentee’s behavior.

---

> ### Author Response · Authors · 2024-11-30
> **Round 2 - Reviewer K1t1 (Part 2)**
>
> 5.**In addition to the above discussion supporting our points on mentor vulnerability, Reviewers CqcN and nkCR did not highlight the vulnerability of our mentors as the main issue of our work.**
>
> We would greatly appreciate specific suggestions for experiments to rigorously evaluate mentor vulnerability or guidance on improving the robustness and generalization of our mentor model. We would be happy to do these experiments and include them in the final version.
>
> We will add Fig. R1 and these discussions in the final version.
>
> In addition, we agree with the reviewer that there is considerable room to improve the mentor's error prediction capabilities. **The objective of our work is NOT to design a superior ultimate SuperMentor.** Instead, we take an initial step in using AI to predict another AI’s error and propose a proof-of-concept SuperMentor model. Despite being a relatively simple ViT architecture, it is still quite remarkable to achieve above-chance error prediction performance. We hope our work inspires researchers to explore more sophisticated AI mentors, incorporating advanced architectures or training strategies, to further enhance the ability to predict the correctness of another AI model’s outputs.

---

### Official Review · Reviewer_pVLx · 2024-11-04

**Soundness:** 3
**Presentation:** 3
**Contribution:** 2
**Rating:** 5
**Confidence:** 4

**Summary:**

Focusing on the vulnerability of DNN models regarding the wrongly predicted samples, this work proposes a framework to predict the errors of trained DNNs over (1) in-domain, (2) out-of-domain, and (3) adversarial samples. The framework trains a “mentor” model to minimize the combination of two losses to (i) mimic the mentee’s prediction and (ii) predict the mentee’s correctness.

**Strengths:**

1. This work focuses on an interesting topic of predicting the errors of trained models over in-domain, out-of-domain, and adversarial errors. This is an important task given the vulnerability and nonlinearity of DNNs.
2. The paper is well-written and very easy to follow.
3. Extensive experiments are carried out.

**Weaknesses:**

1. Main concern: The idea of training another DNN model to predict the vulnerability of another DNN model can be risky. The problems that exist in the training process of the mentee may also exist in the training process of the mentor. For example, what if the backbone of the mentor model is already suffering from these problems?
2. The motivations of the two loss terms are contradicting. $L_d$ encourages the mentor to mimic the mentee’s predictions, including wrong ones. However, $L_r$ encourages the mentor to distinguish the wrong predictions from the correct predictions.
3. The out-of-distribution genre studied in this work is very limited. In this work, only synthetic corruptions such as noise, blur, etc. are included, while important natural distribution shifts such as spurious correlations, styles shift, etc. are completely overlooked. This significantly undermines the real-world contribution of the experimental results.
4. It is observed in the experiments that adversarially trained mentor performs better. However, this could be the benefit of adversarial training as it leads to a smoother and more robust model. If the mentee model is already trained in the adversarial way, the testing accuracy under OOD, and AA could already be improved (e.g. [1]).
5. Due to the aforementioned issues, the contributions of this work remain unclear. It is not straightforward how real-world applications can benefit from this framework. The authors may consider elaborating more on the “high-stakes real-world applications” claimed in L539.


[Minor]

1. Figure 1 can be misleading. Since the authors focus only on the synthetic out-of-distribution samples, the “out-of-distribution images” (middle) look like an adversarial image. Furthermore, the “adversarial images” (bottom) show an attacked sample whose raw image differs from the other two images. This can be easily misunderstood as the out-of-distribution sample with a natural distribution shift. It is strongly suggested that the same raw input be used.
2. The specific formulation of the distillation loss $Distill$ should be given.

[Reference]

[1] Kireev, K., Andriushchenko, M., & Flammarion, N. (2022, August). On the effectiveness of adversarial training against common corruptions. In *Uncertainty in Artificial Intelligence* (pp. 1012-1021). PMLR.

**Questions:**

1. Built also by DNNs, the mentor shall suffer the same problems such as the black-box nature, the high nonlinearity, etc. Does this require another mentor’s mentor to predict whether the mentor can correctly predict the mentee’s prediction? How does the author justify this cyclic scenario?
2. If the mentor can be able to predict the correctness of the mentor with higher accuracy than reconstructing the mentee’s prediction accurately, does this mean that the backbone of the mentor is capable of correctly predicting the true classes of those true negative samples (wrongly predicted by the mentee but recognized by the mentor)? The representations learned from the backbone of the mentor should be an interesting direction to explore.
3. The logistic regression loss $L_r$ is for the binary classification of whether the mentee makes the correct classification. This can be a very imbalanced task. How is this affecting the results?
4. In Figure 2, what is the “average accuracy”? Is it the average of the testing accuracy of the mentor with three ID, OOD, and AA?

---

> ### Author Response · Authors · 2024-11-23
> **Response to the Reviewer pVLx (Part 1)**
>
> **[pVLx.1-Weakness] Main concern: The idea of training another DNN model to predict the vulnerability of another DNN model can be risky. The problems that exist in the training process of the mentee may also exist in the training process of the mentor. For example, what if the backbone of the mentor model is already suffering from these problems?**
>
> Thank you for raising an important point about the potential for shared vulnerabilities between the mentor and mentee models. This is a critical aspect to consider when designing systems that rely on one deep neural network (DNN) to evaluate another. However, we respectfully disagree with the reviewer that both mentees and mentors share the same vulnerability due to the following reasons:
>
> First, in our approach, we intentionally design the mentor model to specialize in detecting and quantifying vulnerabilities of the mentee, rather than solving the primary task of the mentee. This difference in objective between mentors and mentees helps mitigate the risk of overlapping weaknesses. For example, while the mentee might struggle with generalization in a classification task, the mentor focuses solely on identifying patterns indicative of vulnerability, which is a different domain.
>
> Second, the mentor model is designed with robustness in mind. Despite zero training on many error sources in the main text as well as the newly added challenging OOD domains in the rebuttal (see “**Common.2: SuperMentor performance on additional OOD domains on ImageNet dataset**”), our mentor achieves above-chance performance in detecting errors of the mentees. Moreover, we introduce three competitive baselines and our mentor surpasses the performance of these mentor baselines. (see “**Common.1: Performance comparison of mentor baselines and our SuperMentor**”).
>
> Third, the mentor model is designed to be model-agnostic in its evaluations, meaning that it operates independently of the specific architecture or training details of the mentee. This reduces the risk of the mentor inheriting the mentee's specific weaknesses.
>
> Fourth, interestingly, if the mentor model does exhibit similar vulnerabilities, this can serve as an insightful diagnostic tool. Identifying such shared vulnerabilities can highlight systemic issues in the broader DNN training paradigm, offering opportunities to address them at a foundational level. Our framework provides a unique opportunity to improve our understanding of both individual and systemic vulnerabilities in DNNs, leading to more robust AI systems overall. One possible direction is to use GradCAM [h] to visualize the error patterns detected by our mentor at the pixel levels. We will add these points in the discussion section in the final version. Moreover, we will conduct a more rigorous analysis of the error patterns discovered by our mentor in future work.
>
>
> **[pVLx.2-Weakness] The motivations of the two loss terms are contradicting. Ld encourages the mentor to mimic the mentee’s predictions, including wrong ones. However, Lr encourages the mentor to distinguish the wrong predictions from the correct predictions.**
>
> We respectfully disagree with the reviewer on the contradiction of the two introduced losses in the mentor models. For the mentor to make accurate predictions, it must first have a deep understanding of the mentee's behaviors, regardless of the correctness of their responses. Thus, $L_d$ encourages the mentor to mimic the mentee’s predictions. Next, after gaining a deep understanding of the mentee, the mentor uses another MLP branch to predict the correctness of the mentee’s responses with $L_r$ loss. We elaborate on the implementation details of the two losses below:
>
> The mentor has TWO MLP branches, each supervised by different loss terms to perform separate tasks. Each of the MLP branches has separate, non-shared parameters. The mentor's backbone captures the mentee's learning patterns, encompassing both correct and incorrect patterns. The MLP branch associated with loss $L_d$ enables the mentor to mimic the mentee's predictions, and the MLP branch associated with loss $L_r$ works to distinguish between the mentee's error patterns and correct patterns. In other words, the two MLP branches function in parallel, both relying on the mentee’s information captured by the mentor's architecture. Therefore, these two branches are correlated but not contradictory.
>
> We will add these discussions in the final version.

---

> ### Author Response · Authors · 2024-11-23
> **Response to the Reviewer pVLx (Part 2)**
>
> **[pVLx.3-Weakness] The out-of-distribution genre studied in this work is very limited. In this work, only synthetic corruptions such as noise, blur, etc. are included, while important natural distribution shifts such as spurious correlations, styles shift, etc. are completely overlooked. This significantly undermines the real-world contribution of the experimental results.**
>
> That is indeed a very good point! Thank you! As suggested by the reviewer, we have added experiments to evaluate the SuperMentor on more OOD domains in the ImageNet dataset. See “**Common.2: SuperMentor performance on additional OOD domains on ImageNet dataset**”.
>
> To demonstrate the usefulness of our framework in real-world practice, we further expanded our experiments to a medical image classification task, as described in the general comment to all Reviewers “**Common.3: SuperMentor in real-world practice, such as NCTCRCHE100K dataset**”.
>
> We will add these new experimental results in the final version.
>
> **[pVLx.4-Weakness] It is observed in the experiments that adversarially trained mentor performs better. However, this could be the benefit of adversarial training as it leads to a smoother and more robust model. If the mentee model is already trained in the adversarial way, the testing accuracy under OOD, and AA could already be improved (e.g. [1]).**
>
> We would like to clarify that neither our mentor nor our mentee went through any adversarial training at all.
>
> First, the mentor is trained on the mentee's adversarial images rather than its own, which differentiates our approach from adversarial training methods.
>
> Second, our work focuses on improving the mentor's ability to predict errors, rather than enhancing the mentee's testing accuracy under ID, OOD, and AA scenarios. Unlike adversarial training, which involves using adversarial samples to train the mentee, our approach focuses on teaching mentors to learn the mentee’s error patterns revealed by these adversarial attack samples.
>
> We will clarify these points in the final version.
>
> **[pVLx.5-Weakness] Due to the aforementioned issues, the contributions of this work remain unclear. It is not straightforward how real-world applications can benefit from this framework. The authors may consider elaborating more on the “high-stakes real-world applications” claimed in L539.**
>
> As mentioned in **[pVLx.3-Weakness]**, to demonstrate the usefulness of our model in real-world applications, we expanded our experiments to a medical image classification task, as described in the general comment to all Reviewers “**Common.3: SuperMentor in real-world practice, such as NCTCRCHE100K dataset**”.
>
> We will add these new experimental results in the final version.
>
> **[pVLx.6- Minor Weakness] Figure 1 can be misleading. Since the authors focus only on the synthetic out-of-distribution samples, the “out-of-distribution images” (middle) look like an adversarial image. Furthermore, the “adversarial images” (bottom) show an attacked sample whose raw image differs from the other two images. This can be easily misunderstood as the out-of-distribution sample with a natural distribution shift. It is strongly suggested that the same raw input be used.**
>
> Thank you! We will revise the designs of Figure 1 accordingly in the final version.
>
> **[pVLx.7- Minor Weakness] The specific formulation of the distillation loss Distill should be given.**
>
> The distillation loss formula is:
> $L_d = KL(\sigma(\frac{z_E}{T})||\sigma(\frac{z_R}{T}))$, where $KL(\cdot||\cdot)$ represents Kullback-Leibler divergence, $\sigma(\cdot)$ is the softmax function and $T$ is the temperature.
>
> We will add this formula in the final version.

---

> ### Author Response · Authors · 2024-11-23
> **Response to the Reviewer pVLx (Part 3)**
>
> **[pVLx.8- Questions] Built also by DNNs, the mentor shall suffer the same problems such as the black-box nature, the high nonlinearity, etc. Does this require another mentor’s mentor to predict whether the mentor can correctly predict the mentee’s prediction? How does the author justify this cyclic scenario?**
>
> This is an excellent question! However, we argue that these challenges will not hinder the deployment of our framework for error prediction. On the contrary, we view them as an opportunity to introduce a novel framework aimed at enhancing our understanding of DNNs using DNNs themselves.
>
> As the reviewer correctly pointed out, all deep neural networks (DNNs) share common challenges, such as lack of explainability and high nonlinearity. However, these challenges have not deterred researchers from studying DNNs or applying them in real-world scenarios. For example, AlphaFold [i] has revolutionized protein structure prediction, and DNNs are successfully employed in detecting diabetic retinopathy from retinal images [j] —identifying patterns that even doctors had not previously recognized.
> One advantage of these DNN models is that we no longer need to manually design hand-crafted features for downstream tasks. Instead, DNNs automatically discover these patterns for us.
>
> Similarly, we propose the interesting research question of using one DNN to automatically discover the error patterns of the other DNN. Our framework provides a unique opportunity to improve our understanding of both individual and systemic vulnerabilities in DNNs, leading to more robust AI systems overall. One possible direction is to use GradCAM [h] to visualize the error patterns detected by our mentor at the pixel levels. We will add these points in the discussion section in the final version. Moreover, we will conduct a more rigorous analysis of the error patterns discovered by our mentor in future work.
>
> Moreover, as AI models become more integrated into our daily lives, ensuring their correctness is of paramount importance, especially in high-stakes fields such as the medical and finance fields. To demonstrate the usefulness of our mentor model in real-world applications, we expanded our experiments to a medical image classification task, as described in the general comment to all Reviewers “**Common.3: SuperMentor in real-world practice, such as NCTCRCHE100K dataset**”. We will add these experiments in the final version.
>
> **[pVLx.9- Questions] If the mentor can be able to predict the correctness of the mentor with higher accuracy than reconstructing the mentee’s prediction accurately, does this mean that the backbone of the mentor is capable of correctly predicting the true classes of those true negative samples (wrongly predicted by the mentee but recognized by the mentor)? The representations learned from the backbone of the mentor should be an interesting direction to explore.**
>
> This is another brilliant point raised by the reviewer! Thank you!
>
> Ideally, a highly accurate mentor in predicting a mentee's errors can provide valuable insights to enhance the mentee's classification performance. To validate this idea, as suggested by the reviewer, we conducted an experiment where we concatenated the mentee's embeddings (obtained before its final layer for classification) with the trained SuperMentor's embeddings (obtained before its final layer for error prediction). This concatenated representation was passed through a linear layer for the mentee’s final 10-class classification on CIFAR-10, and the mentee was subsequently fine-tuned.
>
> Experimental results demonstrate that the mentee's average classification accuracy across all ID, OOD, and AA testing sets on the CIFAR-10 dataset increases from 90.37% to 92.49%. This indicates that the information learned by the mentor effectively guides the mentee to learn more generalized features, resulting in a more robust mentee for image classification.
>
> We will add this experiment and the result discussions in the final version.

---

> ### Author Response · Authors · 2024-11-23
> **Response to the Reviewer pVLx (Part 4)**
>
> **[pVLx.10- Questions] The logistic regression loss Lr is for the binary classification of whether the mentee makes the correct classification. This can be a very imbalanced task. How is this affecting the results?**
>
> We are unsure what the reviewer means by the term "imbalanced task." We would appreciate it if the reviewer could provide more explanation on this point.
>
> We would like to point out that the reported accuracy is calculated by averaging the mentor's accuracies on the samples that the mentee classified correctly and those that the mentee classified incorrectly. In other words, we have taken into account the long-tailed distribution of the binary classification task. For example, if the mentee correctly classified 10 samples and incorrectly classified 90 samples, and the mentor accurately predicts the correctness of 8 out of the 10 correctly classified samples and 9 out of the 90 incorrectly classified samples, the mentor's accuracy would be calculated as (8/10+9/90)/2 = 45%. We will clarify this point in the final version.
>
> **[pVLx.11- Questions] In Figure 2, what is the “average accuracy”? Is it the average of the testing accuracy of the mentor with three ID, OOD, and AA?**
>
> We could not find the term "average accuracy" in Figure 2; perhaps the reviewer is referring to the "average accuracy" in Figure 3.
>
> This term represents the average accuracy of a mentor across all test sets, including one ID error, four OOD errors, and four AA errors. For instance, a mentor is first trained on the mentee’s predictions from the PIFGSM error source on the CIFAR-10 (C10) dataset. Next, this mentor is evaluated on all the following datasets with their accuracies of 10% on C10-ID, 20% on C10-OOD-SpN, 30% on C10-OOD-GaB, 40% on C10-OOD-Spat, 50% on C10-OOD-Sat, 60% on C10-AA-Jitter, 70% on C10-AA-PGD, 80% on C10-AA-CW, and 90% on C10-AA-PIFGSM testing samples (refer to Sec. 3.3 and Tab. 1 for training and testing splits). Consequently, the average accuracy of this mentor is calculated as (10% + 20% + 30% + 40% + 50% + 60% + 70% + 80% + 90%) / 9 = 50%.
>
> We will add this example to Sec.3.3 and clarify this point in the final version.

---

> ### Comment · Reviewer_pVLx · 2024-11-24
> **Thank the Authors for the Response**
>
> I have read the response from the authors and greatly appreciate the effort in addressing the questions. Most of my questions have been resolved and I have raised the soundness to 3 and the rating to 5 accordingly. However, I stand by my main concern that using another *vulnerable* DNN as the mentor can be a risky operation and deviates from the essence of reliability.

---

> ### Author Response · Authors · 2024-11-25
> **Thank you. We respectfully disagree with the reviewer on the concern regarding vulnerability.**
>
> We respectfully disagree with the reviewer for penalizing us on challenges of vulnerability common to all DNNs. We present the following arguments:
>
> **1.Shared vulnerabilities across all DNNs do not stop us from deploying them in the real-world applications.**
>
> Vulnerability is a well-known challenge for all DNN-based systems, yet this has not hindered their transformative applications in the real world. For example, AlphaFold [i] revolutionized protein structure prediction, and DNNs are used to detect diabetic retinopathy [j], outperforming traditional diagnostic methods. Similarly, our work uses one AI to diagnose the other AI, which is a valid and impactful use case. To demonstrate real-world utility, we extended our experiments to a medical image classification task, as detailed in **Common.3.** Using the NCTCRCHE100K dataset for colorectal cancer diagnosis, our mentor model achieves high classification accuracy, showcasing its practical uses in high-stake applications.
>
> **2.Our AI mentors are robust to error patterns of mentees even in OOD domains that the AI mentors have never been trained on.**
>
> We rigorously evaluate our mentor models across **13 OOD domains** (8 in the main text and 5 newly added in the rebuttal). Despite no prior exposure to these domains, the mentor generalizes well, identifying mentee errors far above chance and surpassing all the competitive baselines. Please refer to **Common.2** and **Common.1** for detailed performance comparisons with all the competitive baselines.
>
> **3. The vulnerabilities of the mentor AI do not overlap with those of the mentees. The mentor's vulnerabilities are less of a concern, as our ultimate goal is to address and eliminate the vulnerabilities of the mentees.**
>
> The vulnerabilities of mentors and mentees are distinct due to their non-overlapping objectives. While mentees may struggle with generalization in specific image classification tasks, mentors are designed to focus solely on error pattern detection of the mentees. Furthermore, mentors operate independently of the mentee’s architecture or training details, reducing the risk of shared weaknesses. This distinction ensures that mentors effectively reduce the vulnerabilities of mentees that matter most in real-world applications.
>
> **4. Our AI mentor serves as a valuable diagnostic tool for the AI mentees.**
>
> Beyond error detection, mentors serve as valuable diagnostic tools of AI mentees. Our framework improves the understanding of both individual and systemic vulnerabilities in DNN-based mentees, fostering more robust AI systems. For instance, GradCAM [h] could visualize the mentor-detected error patterns at the pixel level, offering valuable insights.

---

> ### Author Response · Authors · 2024-11-30
> **Round 2 - Reviewer pVLx**
>
> We thank the reviewer for the initial response. We would like to ask whether the reviewer agrees with our response regarding the vulnerability.
>
> In order to further support our points above, we use EigenCAM [o,p] to visualize the behavior of both our ViT SuperMentor and ViT mentee on example images, as depicted in Fig. R1 at the link [**HERE**](https://drive.google.com/file/d/1DA9yEQ4bf_4YdOjk9ds408Nwet6ERfX5/view?usp=sharing). From these visualizations, we can make two observations below:
>
> 1. **The vulnerabilities of the mentee do not overlap with those of the mentors.** As illustrated in Fig. R1, the mentor does not merely replicate the mentee’s learning patterns, as their activation maps for identical input images can differ significantly. For example, in the images shown in Fig. R1 (a), although the mentor and mentee exhibit very similar activation maps for ID images, their activations diverge greatly when the input image is corrupted by Gaussian Blur (GaB) or subjected to adversarial attacks using PIFGSM method. In the IN-GaB and IN-PIFGSM images, the mentee demonstrates vulnerability under both corruption and adversarial attacks, whereas our SuperMentor does not exhibit such vulnerabilities on these OOD and AA images, consistently focusing on the brambling object. This indicates the vulnerabilities of mentors and mentees are distinct due to their non-overlapping objectives.
>
> 2. **Our AI mentor serves as a valuable diagnostic tool for AI mentees.** We argue that, beyond merely detecting errors, mentors function as important diagnostic tools for AI mentees, as supported in Fig. R1. For example, in Fig. R1(b), although the mentee correctly classifies the ID image, its activation map does not concentrate on the baseball but instead highlights background cues (spurious features). When the input image is corrupted by Gaussian Blur (GaB) or subjected to adversarial PIFGSM attacks, the mentee fails to make accurate predictions. In contrast, our SuperMentor effectively
> focuses on the stitches of the baseball—a key feature—regardless of whether the image is corrupted or adversarially attacked. The mentor highlights the stitches of the baseball as the key feature pattern that the mentee tends to overlook. This suggests that the mentor can identify vulnerabilities in the DNN-based model, even though it shares the same ViT architecture as the mentee. Furthermore, this reveals significant potential for our mentor to help refine the mentee’s performance by providing valuable insights into the mentee’s behavior.
>
> In conclusion, Fig. R1 provides strong evidence that the AI mentor not only does not share the same vulnerabilities as the mentee but also serves as a valuable diagnostic tool for AI mentees.
>
> **In addition to the above discussion supporting our points on mentor vulnerability, Reviewers CqcN and nkCR did not highlight the vulnerability of our mentors as the main issue of our work.**
>
> We would greatly appreciate specific suggestions for experiments to rigorously evaluate mentor vulnerability or guidance on improving the robustness and generalization of our mentor model. We would be happy to do these experiments and include them in the final version.
>
> We will add Fig. R1 and these discussions in the final version.

---

### Official Review · Reviewer_CqcN · 2024-11-04

**Soundness:** 3
**Presentation:** 3
**Contribution:** 2
**Rating:** 6
**Confidence:** 3

**Summary:**

The work uses a dedicated neural network to detect 3 classes of error sources
in image classification, including adversarial examples, image distortion, and
in-distribution prediction error. The detection model is analyzed in various
ablation studies, including the effect of the strength of the different error
sources in accuracy, the accuracy under different architectures for both
detector and predictor (ResNet and ViT), and the distillation loss. A t-SNE
embedding is shown for the features of the detector.

**Strengths:**

- The work is well written and clearly presents its contributions.

- The work introduces a novel comparison for detection difficulty between
  different error sources in image classification.

- Various ablation studies are conducted for the proposed model.

**Weaknesses:**

- Novelty: Adversarial Examples are an extremely well researched topic. (see,
  e.g., Yuan et al., 2019) for a survey). Hence, the idea of detecting
  adversarial examples by using a classifier has been done before (see, e.g., Metzen
  et al., 2017). This limits the novel contribution of this work to the
  comparison of the detection of the different error types.

- Contribution: Although the work references other approaches for the detection of
  adversarial attacks and other presented error sources, there is no baseline
  comparison at all. For a significant contribution, it is necessary to compare
  it to previous work which attempts to solve the same issue. The work reports
  some accuracy of up to 83%, yet it is difficult to quantify this without
  analyzing the detection accuracy achieved through other approaches.

- Clarity: The work presents multiple ablation studies to compare how different
  architectures perform under the discussed error sources. While interesting,
  it is not quite clear to me why this is relevant unless compared to other
  baselines. I also did not understand how the t-SNE embeddings support the
  claim that the proposed neural network-based detector is a good idea.

In total, this work requires a more thorough literature search concerning
different approaches to errors in image classification. It is very difficult to
justify a method when it is not compared to previous work.


References:

X. Yuan, P. He, Q. Zhu and X. Li, "Adversarial Examples: Attacks and Defenses
for Deep Learning," in IEEE Transactions on Neural Networks and Learning
Systems, vol. 30, no. 9, pp. 2805-2824, Sept. 2019.

Metzen, J. H., Genewein, T., Fischer, V., & Bischoff, B. . "On
Detecting Adversarial Perturbations". ICLR 2017.

**Questions:**

- Did you compare your approach to other baselines to detect model error?

- How does your approach differ from (Metzen et al., 2017), except for the different error types?

- What purpose do the t-SNE embeddings serve?

---

> ### Author Response · Authors · 2024-11-23
> **Response to the Reviewer CqcN (Part 1)**
>
> **[CqcN.1-Weakness] Novelty: Adversarial Examples are an extremely well researched topic. (see, e.g., Yuan et al., 2019) for a survey). Hence, the idea of detecting adversarial examples by using a classifier has been done before (see, e.g., Metzen et al., 2017). This limits the novel contribution of this work to the comparison of the detection of the different error types.**
>
> We respectfully disagree with the reviewer. Our work is NOT about adversarial attacks or the detection of adversarial attacks. Instead, our aim is to develop a mentor that can accurately predict the correctness of a mentee’s performance. Specifically, we explore whether the mentor can predict a mentee's performance on different types of errors. We hope this explanation clarifies the problem setting of our work.
>
> **[CqcN.2-Weakness] Contribution: Although the work references other approaches for the detection of adversarial attacks and other presented error sources, there is no baseline comparison at all. For a significant contribution, it is necessary to compare it to previous work which attempts to solve the same issue. The work reports some accuracy of up to 83%, yet it is difficult to quantify this without analyzing the detection accuracy achieved through other approaches.**
>
> First, our work is NOT about adversarial attacks or the detection of adversarial attacks. Instead, our aim is to develop a mentor that can accurately predict the correctness of a mentee’s prediction. Specifically, we explore whether the mentor can predict a mentee's performance on different types of errors. We hope this explanation clarifies the problem setting of our work.
>
> Second, to the best of our knowledge, our study is the first to use one AI model to predict the correctness of another AI model’s predictions. Due to the lack of existing AI mentors, we have introduced three baseline approaches based on the mentee’s confidence, the entropy of the mentee’s logits, and the L2 distance between the mentee’s features and the class feature centroids, respectively. The details of experimental results are described in the general comment to all Reviewers “**Common.1: Performance comparison of mentor baselines and our SuperMentor**”.
>
> We will add these new experimental results in the final version.
>
> **[CqcN.3-Weakness] Clarity: The work presents multiple ablation studies to compare how different architectures perform under the discussed error sources. While interesting, it is not quite clear to me why this is relevant unless compared to other baselines. I also did not understand how the t-SNE embeddings support the claim that the proposed neural network-based detector is a good idea.**
>
> First, as required by the reviewer, we now provide our own baselines for comparisons. See **[CqcN.2-Weakness]** above.
>
> Second, we use t-SNE to perform clusterings on the representations of our SuperMentor model for correctness classifications of the mentees based on different error sources from the CIFAR-10 dataset. This visualization demonstrates the mentor's capability to predict the correctness of the mentee’s performance. It is worth noting that our work introduces a binary classification task instead of an error monitoring task on classifying the different types of error sources. We will clarify this question and make our figure captions clearer in the final version.

---

> ### Author Response · Authors · 2024-11-23
> **Response to the Reviewer CqcN (Part 2)**
>
> **[CqcN.4-Question] Did you compare your approach to other baselines to detect model error?**
>
> To the best of our knowledge, our study is the first to use one AI model to predict the correctness of another AI model’s predictions. Due to the lack of existing AI model baselines, we have introduced three baseline approaches based on the mentee’s confidence, the entropy of the mentee’s logits, and the L2 distance between the mentee’s features and the class feature centroids, respectively. The details of experimental results are described in the general comment to all Reviewers “**Common.1: Performance comparison of mentor baselines and our SuperMentor**”.
>
> We will add these new experimental results in the final version
>
> **[CqcN.5-Question] How does your approach differ from (Metzen et al., 2017), except for the different error types?**
>
> Our work is NOT about adversarial attacks or the detection of adversarial attacks. Instead, our aim is to develop a mentor that can accurately predict the correctness of a mentee’s performance regardless of what type of mistakes the mentee makes. Specifically, we explore whether the mentor can predict a mentee's performance on different types of errors. We hope this explanation clarifies the problem setting of our work.
>
> **[CqcN.6-Question] What purpose do the t-SNE embeddings serve?**
>
> We use t-SNE to perform clusterings on the representations of our SuperMentor model for classifications of different error sources on the CIFAR-10 dataset. This visualization demonstrates the mentor's capability to predict the correctness of the mentee’s performance. It is worth noting that our work introduces a binary classification task instead of an error source classification task.

---

> > ### Comment · Reviewer_CqcN · 2024-11-25
> >
> > Thank you for your clarifications.
> > I highly appreciate the added baseline experiments, and the extensive comments.
> >
> > I acknowledge that the experiments include two perturbation (error)
> > types different from adversarial examples, and I am also aware that
> > the mentor is trained to classify the prediction error of the model *on* the
> > different error types, rather than detecting the presence of the perturbation
> > in the data itself.
> >
> >
> > ## Misclassification Detection
> >
> > While I was not fully familiar with the field, it seems more appropriate to
> > view this manuscript in the context of "Misclassification Detection",
> > "Trustworthiness Prediction", or partially "Confidence Estimation", without
> > entering the field of uncertainty estimation too much. The baselines for this
> > work, as now conducted in the updated version, usually depend on the confidence
> > scores of the network (as provided in the cited (Hendryks and Gimpel, 2017)).
> >
> > Here, fairly related work seems to be (Corbière et al., 2019), in which a
> > separate network "ConfidNet" is trained on network features to predict a "true
> > confidence probability", which indicates the misclassification. This approach is
> > further refined in (Luo et al, 2021).
> > Qiu et al. 2022 use instead a Gaussian Process to identify misclassification.
> >
> >
> > ## Discussion
> >
> > I would really like to see the manuscript put itself more in context of similar
> > work. While I appreciate the basic baselines, I see no reason why approaches
> > such as Trust Score (Jiang et al., 2019) could not be compared as a baseline.
> >
> > I think that in general, the paper is in a good shape, and the experiments are
> > well conducted, where the split of the error types is especially interesting
> > for the analysis.
> > Given, however, the missing context, prior work, and especially the fact that a
> > "neural network"-based misclassification detection has in fact been done before,
> > I somewhat worry about the novelty of the approach itself.
> >
> > In the current state, I am confident to increase my score to 6, provided that
> > the manuscript is put more into context with the works discussed above. While I
> > understand that time is limited, I do not feel confident to increase my vote
> > beyond 6 without the empirical comparison to prior work such as "ConfidNet" and "Trust
> > Score".
> >
> >
> > ## References:
> >
> > Dan Hendrycks, & Kevin Gimpel (2017). A Baseline for Detecting Misclassified
> > and Out-of-Distribution Examples in Neural Networks. In International
> > Conference on Learning Representations.
> >
> > Corbière, C., Thome, N., Bar-Hen, A., Cord, M., & Pérez, P. (2019). Addressing
> > failure prediction by learning model confidence. Advances in Neural Information
> > Processing Systems, 32.
> >
> > Luo, Y., Wong, Y., Kankanhalli, M. S., & Zhao, Q. (2021). Learning to predict
> > trustworthiness with steep slope loss. Advances in Neural Information
> > Processing Systems, 34, 21533-21544.
> >
> > Qiu, X., & Miikkulainen, R. (2022, June). Detecting misclassification errors in
> > neural networks with a gaussian process model. In Proceedings of the AAAI
> > Conference on Artificial Intelligence (Vol. 36, No. 7, pp. 8017-8027).
> >
> > Jiang, H., Kim, B., Guan, M., & Gupta, M. (2018). To trust or not to trust a
> > classifier. Advances in neural information processing systems, 31.

---

> > > ### Author Response · Authors · 2024-11-30
> > > **Round 2 - Reviewer CqcN (Part 1)**
> > >
> > > Thank you for highlighting these amazing works [k-n]. They share a similar focus with our work, as all of them aim to address the topic of trustworthy AI. However, there are several key differences between our work and their work, which are outlined below:
> > >
> > > **Addressing failure prediction by learning model confidence [k]:**
> > >
> > > [k] discovered that True Class Probability (TCP) is more effective for error prediction compared to the traditional Maximum Class Probability (MCP). It introduced a model called ConfidNet to predict the confidence score of mentees. Our work differs from theirs in two key aspects:
> > >
> > > 1) The proposed ConfidNet is trained to estimate the ground truth confidence score, corresponding to the mentee's true class probability. In contrast, our mentor mimics the mentee's predictions using a distillation loss. The distillation loss not only captures the true class information but also accounts for the false class information, enabling the mentor to develop a more comprehensive understanding of the mentee's behavior.
> > >
> > > 2) The proposed ConfidNet is a shallow neural network consisting of a series of layers, whereas our mentor employs deeper architectures like ResNet50 and ViT, allowing it to capture more intrinsic error patterns in the mentee's predictions.
> > >
> > > **Learning to predict trustworthiness with steep slope loss [l]:**
> > >
> > > The oracle in [l] is trained using the mentee's training samples, whereas in our approach, the mentee's training data is inaccessible. Our SuperMentor is trained only on the mentee's performance on adversarial images.  Our work demonstrates that adversarial images, which lie closer to the mentee’s decision boundary, enable more precise predictions of the mentee’s errors and offer a deeper insight into the mentee's loss landscape. As a result, our training strategy becomes more efficient because the AA data used effectively captures the mentee’s learning patterns.
> > >
> > > **Detecting misclassification errors in neural networks with a gaussian process model [m]:**
> > >
> > > The work in [m] presents a new framework called RED to detect errors on top of the base classifier and estimates the uncertainty of the detection scores using Gaussian Processes. Our work mainly differs from theirs in one aspect: The mRIO model in [m] is not a deep neural network (DNN), whereas the mentors in our work are DNNs. This distinction allows our mentor to effectively learn and adapt to the mentee's complex learning patterns. In contrast, the approach in [m] may struggle to capture and leverage such intricate patterns for error prediction.
> > >
> > > **To trust or not to trust a classifier [n]:**
> > >
> > > [n] introduces a new score called trust score by measuring the agreement between the classifier and a modified nearest-neighbor classifier on the testing example. Our work differs from theirs in two key aspects:
> > >
> > > 1) The nearest-neighbor classifier may not align well with the true error patterns of the mentee. In contrast, our mentor is an AI model specifically designed to learn and understand the behaviors of mentees. This enables it to make error predictions based on these behaviors, resulting in more robust, efficient, and informed decision-making.
> > >
> > > 2) The trust score is not derived from a deep neural network (DNN), whereas the mentors in our approach are implemented as DNNs. This distinction allows our mentor to effectively learn and adapt to the mentee's complex learning patterns. In contrast, the approach in [n] may struggle to capture and leverage such intricate patterns for error prediction.
> > >
> > > Moreover, we also implemented their methods and conducted fair comparisons in performance between their methods with ours. The results are presented in Tab. R5 and R6. It clearly shows that SuperMentor achieves higher average accuracies across all error types compared to the baselines. This demonstrates that the source of errors used to train the mentor can be more critical than the choice of training loss functions or strategies in those baselines. Despite the baselines employing sophisticated loss functions and training strategies, their performance is surpassed by our SuperMentor, which is trained using AA data only.
> > >
> > > We will cite these works, and add these discussions and experiments in the final version.

---

> > > ### Author Response · Authors · 2024-11-30
> > > **Round 2 - Reviewer CqcN (Part 2)**
> > >
> > > **Table R5: Performance comparison of mentor baselines and our SuperMentor on the ImageNet dataset. The ResNet50 model serves as the mentee, with its correctness of predictions evaluated by mentors. The three baseline methods are ConfidNet [k], TrustScore [n] and Steep Slope Loss (SS) [l] respectively. Best results are in bold.**
> > > |                    |  ID  |  SpN |  GaB | Spat |  Sat |  PGD |  CW  | Jitter | PIFGSM | Average |
> > > |--------------------|:----:|:----:|:----:|:----:|:----:|:----:|:----:|:------:|:------:|:-------:|
> > > |      ConfidNet     | 67.3 | 65.9 | 68.5 | 70.8 | 67.9 | 68.8 | 69.0 |  70.1  |  67.1  |   68.4  |
> > > |      TrustScore    | 70.1 | 67.6 | 71.1 | 73.4 | 71.3 | 73.3 | 72.6 |  73.9  |  72.7  |   71.8  |
> > > |         SS         | 66.3 | 66.7 | 69.7 | 71.9 | 70.0 |69.1 | 70.1 |  70.2  |  66.7  |  69.0  |
> > > | SuperMentor (ours) | **78.9** | **73.6** | **74.6** | **78.3** | **73.6** | **83.0** | **78.4** |  **79.9**  |  **72.3**  |   **77.0**  |
> > >
> > > **Table R6: Performance comparison of mentor baselines and our SuperMentor on the ImageNet dataset. The ViT model serves as the mentee, with its correctness of predictions evaluated by mentors. The three baseline methods are ConfidNet [k], TrustScore [n] and Steep Slope Loss (SS) [l] respectively. Best results are in bold.**
> > >
> > > |                    |  ID  |  SpN |  GaB | Spat |  Sat |  PGD |  CW  | Jitter | PIFGSM | Average |
> > > |--------------------|:----:|:----:|:----:|:----:|:----:|:----:|:----:|:------:|:------:|:-------:|
> > > |      ConfidNet     | 73.3 | **74.1** | **73.4** | **75.6** | **74.5** | 69.2 | 70.5 |  66.9  |  70.5  |   72.0  |
> > > |      TrustScore    | **73.7** | 72.0 | 72.4 | 73.5 | 72.5 | 70.4 | 70.1 |  69.2  |  72.5  |   71.8  |
> > > |         SS         | 64.3 | 66.3 | 68.2 | 70.6 | 69.3 |62.5 | 66.0 |  60.1  | 62.4  |  65.5  |
> > > | SuperMentor (ours) | 72.7 | 71.3 | 71.7 | 73.0 | 70.2 | **80.2** | **74.1** |  **72.9**  |  **74.1**  |   **73.4**  |

---

### Author Response · Authors · 2024-11-23
**Common.1: Performance comparison of mentor baselines and our SuperMentor (Part 1)**

To the best of our knowledge, our study is the first to use one AI model to predict the correctness of another AI model’s predictions. Due to the lack of existing AI model baselines, we have introduced three baseline approaches based on the mentee’s confidence (Confidence), the entropy of the mentee’s logits (Entropy), and the L2 distance between the mentee’s features and the class feature centroids (Distance to Class Centroids), respectively. The details of these three baseline methods are provided below.

1) Confidence: As mentioned in Sec. 3.1, the mentee’s output logit is denoted as $z_E$. The confidence of the mentee’s output is defined as $Confidence(z_E) = \max\sigma(z_E))$ where $\sigma(\cdot)$ is the softmax function. The confidence indicates how confident the mentee is in its most probable prediction.  We set a threshold $\gamma$ to distinguish between the mentee's confident and unconfident predictions. Specifically, if $Confidence(z_E) > \gamma$, the mentee’s prediction will be considered a correct prediction due to its high confidence; otherwise, it will be regarded as an incorrect prediction. We employ three predefined thresholds for $\gamma$: 0.5, 0.7, and 0.9.

2) Entropy: The entropy of the mentee’s output is defined as $Entropy(z_E) = H(\sigma(z_E))$, where $\sigma(\cdot)$ denotes the softmax function and $H(\cdot)$ represents the entropy measure quantifying the uncertainty in the probability distribution. A high entropy value signifies a high level of uncertainty in the mentee’s predictions. The entropy reaches its maximum value, $MaxEntropy(z_E)$, when the mentee’s class probabilities in $\sigma(z_E)$ are equal. We define the uncertainty threshold as $\alpha \cdot MaxEntropy(z_E)$, where $\alpha \in [0,1]$. If $Entropy(z_E) < \alpha \cdot MaxEntropy(z_E)$, the mentee’s prediction is considered correct, indicating sufficient certainty in its prediction. Otherwise, the prediction is regarded as incorrect. We set three predefined values for $\alpha$: 0.01, 0.1 and 0.3.

3) Distance to Class Centroid: The embedding generated by the mentee before the final binary classification layer represents the mentee’s feature interpretation of each sample. To determine the feature centroid for each class, we first average the features of all testing samples within that class based on the mentee's predictions. Next, we calculate the L2 distance between each sample’s feature and its corresponding class centroid, denoted as $d_s$. We establish a distance threshold $d$; if $d_s < d$, the mentee’s prediction is considered correct since the sample is close to the class centroid. Otherwise, the prediction is deemed incorrect. We have set three predefined values for $d$: 10, 20, and 30.

The experimental results for the three baseline methods applied to ResNet50 and ViT mentees are shown in Tab. R1 and Tab. R2, respectively. Our SuperMentor consistently achieves better average accuracies over all error types compared to all baselines. Notably, in the AA scenarios, SuperMentor has significantly higher accuracy than the baselines. For example, SuperMentor achieves an 80.2% prediction accuracy for the correctness of ViT mentee performance on data generated by PGD attack, whereas the best baseline only reaches 63.9%. This implies that the error prediction of AI mentees is complex and it requires more than manually defined criteria such as confidence or entropy.

Additionally, all these baselines have the drawback of requiring predefined values or thresholds, which are not adaptive and are difficult to select for different OOD and AA scenarios. In contrast, our SuperMentor does not have these limitations. Therefore, using AI models for error predictions is more feasible and does not require hand-crafted features.

We will add these new experiments in the final version.

---

### Author Response · Authors · 2024-11-23
**Common.1: Performance comparison of mentor baselines and our SuperMentor (Part 2)**

**Table R1: Performance comparison of mentor baselines and our SuperMentor trained on various types of errors from the ImageNet dataset. The ResNet50 model serves as the mentee, with its correctness of predictions evaluated by mentors. The three types of baselines are based on the mentee’s confidence, the entropy of the mentee’s logits, and the L2 distance between the mentee’s features and the centroids of the class features respectively. Best results are in bold.**

|  | ID | SpN | GaB | Spat | Sat | PGD | CW | Jitter | PIFGSM | **Average** |
|---|:---:|:---:|:---:|:---:|:---:|:---:|:---:|:---:|:---:|:---:|
| Confidence Threshold  $\gamma = 0.5$  | 69.5 | 73.5 | 73.6 | 72.2 | 71.7 | 66.0 | 66.7 | 65.1 | 64.9 | 69.2 |
| Confidence Threshold  $\gamma = 0.7$  | 77.3 | **76.8** | **77.4** | **78.3** | **76.6** | 71.8 | 72.9 | 71.7 | 69.4 | 74.7 |
| Confidence Threshold  $\gamma = 0.9$  | 78.2 | 73.3 | 74.5 | 77.0 | 74.9 | 72.2 | 72.3 | 73.0 | 69.0 | 73.8 |
| Entropy Coefficient $\alpha = 0.01$ | 69.1 | 62.1 | 63.4 | 66.0 | 64.0 | 63.7 | 62.6 | 65.1 | 60.7 | 64.1 |
| Entropy Coefficient $\alpha = 0.1$ | 78.0 | 75.2 | 75.9 | 78.0 | 76.2 | 72.2 | 72.8 | 72.7 | 69.3 | 74.5 |
| Entropy Coefficient $\alpha = 0.3$ | 64.0 | 70.1 | 69.6 | 67.6 | 67.0 | 61.2 | 61.5 | 60.6 | 60.8 | 64.7 |
| Distance Threshold $d = 10$ | 50.1 | 50.2 | 50.0 | 50.5 | 50.5 | 50.2 | 50.1 | 50.0 | 50.2 | 50.2 |
| Distance Threshold $d = 20$ | 46.0 | 45.4 | 44.3 | 47.9 | 46.2 | 50.1 | 48.4 | 46.3 | 47.3 | 46.9 |
| Distance Threshold $d = 30$ | 48.8 | 49.3 | 49.7 | 49.3 | 49.5 | 49.5 | 49.4 | 49.6 | 49.6 | 49.4 |
| SuperMentor (Ours) | **78.9** | 73.6 | 74.6 | 78.3 | 73.6 | **83.0** | **78.4** | **79.9** | **72.3** | **77.0** |

**Table R2: Performance comparison of mentor baselines and our SuperMentor trained on various types of errors from the ImageNet dataset. The ViT model serves as the mentee, with its correctness of predictions evaluated by mentors. The three types of baselines are based on the mentee’s confidence, the entropy of the mentee’s logits, and the L2 distance between the mentee’s features and the centroids of the class features respectively. Best results in bold.**

|  | ID | SpN | GaB | Spat | Sat | PGD | CW | Jitter | PIFGSM | **Average** |
|---|:---:|:---:|:---:|:---:|:---:|:---:|:---:|:---:|:---:|:---:|
| Confidence Threshold  $\gamma = 0.5$  | 69.9 | 72.9 | 71.5 | 71.5 | 71.8 | 60.5 | 63.6 | 56.6 | 60.7 | 66.6 |
| Confidence Threshold  $\gamma = 0.7$  | **78.5** |**77.9** | **78.0** | **78.2** | **78.0** | 63.9 | 67.8 | 61.5 | 63.8 | 71.9 |
| Confidence Threshold  $\gamma = 0.9$  | 58.0 | 53.3 | 57.2 | 55.5 | 57.6 | 50.6 | 51.6 | 50.9 | 49.2 | 53.8 |
| Entropy Coefficient $\alpha = 0.01$ | 50.0 | 50.0 | 50.0 | 50.0 | 50.0 | 50.0 | 50.0 | 50.0 | 50.0 | 50.0 |
| Entropy Coefficient $\alpha = 0.1$ | 51.4 | 50.7 | 51.2 | 51.1 | 51.5 | 49.6 | 50.1 | 49.6 | 49.1 | 50.5 |
| Entropy Coefficient $\alpha = 0.3$ | 72.4 | 73.7 | 73.5 | 72.7 | 73.5 | 61.3 | 63.7 | 57.0 | 60.2 | 67.5 |
| Distance Threshold $d = 10$ | 52.6 | 52.0 | 51.7 | 54.6 | 52.1 | 52.7 | 52.4 | 52.1 | 53.7 | 52.6 |
| Distance Threshold $d = 20$ | 55.6 | 59.1 | 52.8 | 54.6 | 53.2 | 54.8 | 53.1 | 51.3 | 53.4 | 54.2 |
| Distance Threshold $d = 30$ | 50.0 | 50.0 | 50.0 | 50.0 | 50.0 | 50.0 | 50.0 | 50.0 | 50.0 | 50.0 |
| SuperMentor (Ours) | 72.7 | 71.3 | 71.7 | 73.0 | 70.2 | **80.2** | **74.1** | **72.9** | **74.1** | **73.3** |

---

### Author Response · Authors · 2024-11-23
**Common.2: SuperMentor performance on additional OOD domains on ImageNet dataset**

We included experiments to evaluate the SuperMentor in more diversified OOD domains on ImageNet datasets, including ImageNet9-MIXED-RAND (IN9-MR) [a],  ImageNet9-MIXED-SAME (IN9-MS) [a],  ImageNet9-MIXED-NEXT (IN9-MN) [a], ImageNet-R (IN-R) [b] and ImageNet-Sketch (IN-S) [c]. Specifically, the MIXED-RAND, MIXED-SAME, and MIXED-NEXT datasets are derived from 9 classes in ImageNet and contain varying amounts of background and foreground signals. These datasets aim to demonstrate that models often classify objects based on background cues (often spurious features), rather than the objects themselves. Specifically, MIXED-SAME, MIXED-RAND, and MIXED-NEXT represent images with random backgrounds from the same class, random backgrounds from a random class, and random backgrounds from the next class, respectively. The ImageNet-R dataset comprises images featuring artistic representations of objects, such as cartoons, community-generated art, and graffiti renditions. The ImageNet-Sketch dataset consists of sketch-like images that match the ImageNet validation set in both categories and scale.

The experimental results are shown in Tab. R3. It is worth noting that the SuperMentor learns from the mentee's errors on adversarial ImageNet images generated by the PIFGSM attack only. In other words, the SuperMentor was **NOT** trained on any of these additional ImageNet-related OOD datasets. Yet, the SuperMentor achieves prediction accuracy above the 50% chance level. Despite that our Supermentor model is a simple ViT, it is still quite remarkable to achieve above-chance error prediction performance. We hope our work inspires researchers to explore more sophisticated AI mentors, incorporating advanced architectures or loss functions, to further enhance the ability to predict the correctness of another AI model’s outputs.

We will add these new experiments in the final version.

**Table R3: Performance of SuperMentor on additional ImageNet-related OOD datasets. The abbreviations IN9-MR, IN9-MS, IN9-MN, IN-R, and IN-S stand for ImageNet9-MIXED-RAND [a], ImageNet9-MIXED-SAME [a], ImageNet9-MIXED-NEXT [a], ImageNet-R [b], and ImageNet-Sketch [c], respectively. Results denote the average accuracy with over 3 runs.**

|             |    IN9-MR    |    IN9-MS    |    IN9-MN    |     IN-R     |     IN-S     |
|:-----------:|:------------:|:------------:|:------------:|:------------:|:------------:|
| Accuracy(%) | 68.46 | 69.24 | 67.04 | 58.68 | 58.68 |
|             |              |              |              |              |              |

---

### Author Response · Authors · 2024-11-23
**Common.3: SuperMentor in real-world practice, such as NCTCRCHE100K dataset**

To reflect the real-world contribution of our work, we expanded our experiments to the NCTCRCHE100K [g] dataset which is used for medical image classification on colorectal cancer. This dataset comprises around 100K images across 9 tissue classes. Predicting AI models’ errors in medical image classification has significant real-world implications, such as reducing misdiagnoses and increasing the reliability of AI-assisted medical tools, underscoring the real-world contribution of our study. The results presented in Tab. R4 demonstrate that the mentor accurately determines the correctness of up to 81.9 % of the mentee’s predictions compared to the chance level of 50%. The highest accuracy is achieved by training the mentor on the mentee's adversarial images generated by PIFGSM, which aligns with the SuperMentor configurations in our study.

As AI models become more integrated into our daily lives, ensuring their correctness is of paramount importance, especially in high-stakes fields such as the medical and finance fields. Our experiments with medical image datasets provide a compelling example of the critical role our work plays in these fields, demonstrating its significant impact on real-world applications.

We will add these new experiments in the final version.

**Table R4: The average accuracy of mentor models trained on various error sources for the NCTCRCHE100K [g] dataset. The mentor and mentee are ViT and ResNet50 respectively. Results denote the average accuracy over 3 runs.**
|              |      ID     |     SpN    |     GaB     |     Spat    |     Sat     |     PGD     |      CW     |    Jitter   |    PIFGSM   |
|--------------|:-----------:|:----------:|:-----------:|:-----------:|:-----------:|:-----------:|:-----------:|:-----------:|:-----------:|
| NCTCRCHE100K | 77.9 | 73.3 | 76.8 | 81.2  | 64.8 | 80.9 | 79.3 | 81.4 | 81.9 |

---

### Author Response · Authors · 2024-11-23
**Reference List**

References:

[a] Xiao, Kai, et al. Noise or signal: The role of image backgrounds in object recognition. arXiv 2020.

[b] Hendrycks, Dan, et al. The many faces of robustness: A critical analysis of out-of-distribution generalization. ICCV 2021.

[c] Wang, Haohan, et al. Learning robust global representations by penalizing local predictive power. NeurIPS 2019.

[d] Cortes, et al. Learning with rejection. ALT 2016.

[e] Madras, et al. Predict responsibly: improving fairness and accuracy by learning to defer. NeurIPS 2018.

[f]  Geifman, et al. Selective classification for deep neural networks. NeurIPS 2017.

[g] Kather, et al. Predicting survival from colorectal cancer histology slides using deep learning: A retrospective multicenter study. PLoS medicine 2019.

[h] Selvaraju, et al. Grad-cam: Visual explanations from deep networks via gradient-based localization. ICCV 2017.

[i] Jumper, et al. Highly accurate protein structure prediction with AlphaFold. Nature 2021.

[j] Alyoubi, et al. Diabetic retinopathy detection through deep learning techniques: A review. Informatics in Medicine Unlocked, 2020.

[k] Corbière et al. Addressing failure prediction by learning model confidence. NuerIPS 2019.

[l] Luo. Learning to predict trustworthiness with steep slope loss.  NuerIPS 2021.

[m] Qiu et al. Detecting misclassification errors in neural networks with a gaussian process model. AAAI 2022.

[n] Jiang et al. To trust or not to trust a classifier. NuerIPS 2018.

[o] Jacob Gildenblat and contributors. Pytorch library for cam methods. https://github.com/jacobgil/pytorch-grad-cam, 2021.

[p] Mohammed et al. Eigen-cam: Class activation map using principal components. IJCNN, 2020.

---

### Meta-Review · Area_Chair_dL9Q · 2024-12-23

**Metareview:**

This paper proposes a supermentor learning framework with the goal to predict mentee models error in three main settings: in-domain error, out-of-domain error and adversarial attack. While the idea of using DNNs as mentor model and the framework is interesting and that the authors did a great job providing additional experiment results (especially comparing with prior baselines, as suggested by multiple reviewers), there are some unresolved concerns on using another AI model to predict the error of an AI model, such as vulnerability of the mentor model. While the authors clarified that the mentor model's behavior is very different from the mentee models due to the distinct training objectives, a more fundamental question/concern still persists on this line of work: why and when do we want to use another model to predict the error of the model of interest, instead of directly evaluating the error of itself? Based on the setup, the mentor model and the mentee model both have the access to the same input data and the authors assume the white-box access to mentee models, the proposed research seem to be a redundant approach to understand the behavior of the model of interest (i.e. mentee models in this framework). The authors are encouraged to further investigate the practical use case of the proposed approach to justify the importance of the work. Therefore, a rejection is recommended.

**Additional Comments On Reviewer Discussion:**

- There are shared concerns regarding no baselines comparison in the experiments to the proposed methods. In response, the authors provide additional baselines as well as prior work comparison (e.g. confidnet, trustscore, ss), to show its competitive performance. The additional results strengthen the paper and the authors are suggested to include the results in the main draft.

- There are shared concerns on the vulnerabilities of the mentor model (DNN-based) and the applications of this approach. While the authors clarified the behaviors of the mentor models are very different from mentee model and provided additional medical image classification result, a fundamental concern is still remaining for this line of work - when and why do we need to use another AI model predict the model of interest?

- There are questions regarding the difference of this work v.s. other similar/related work in the literature that were not discussed, including misclassification detection, selective prediction, rejection learning. The authors did a good job to clarify the difference between these prior work in the rebuttal.

---

### Decision · Program_Chairs · 2025-01-22

Reject